# Efficient Path Planning for Collision Avoidance of Construction Vibration Robots Based on Euclidean Signed Distance Field and Vector Safety Flight Corridors

**DOI:** 10.3390/s25061765

**Published:** 2025-03-12

**Authors:** Lei Li, Lingjie Kong, Chong Liu, Hong Wang, Mingyang Wang, Dongxu Pan, Jiasheng Tan, Wanji Yan, Yiang Sun

**Affiliations:** 1College of Mechanical Engineering and Automation, Northeastern University, Shenyang 110819, China; 2410101@stu.neu.edu.cn (L.L.); congliu@me.neu.edu.cn (C.L.); 2270221@stu.neu.edu.cn (M.W.); 2300473@stu.neu.edu.cn (W.Y.); 2400409@stu.neu.edu.cn (Y.S.); 2China Construction Eighth Engineering Division Corp., Ltd., Shanghai 200112, China; lingjiekong@mail.dlut.edu.cn (L.K.); 15840568689@163.com (D.P.); 1219tan@163.com (J.T.)

**Keywords:** vibration robots, autonomous navigation, safe flight corridors, ESDF, motion palnning

## Abstract

Traditional manual concrete vibration work faces numerous limitations, necessitating the need for efficient automated methods to support this task. This study proposes a path safety optimization method based on safe flight corridors and Euclidean signed distance fields, which is suitable for flexible autonomous movement of vibrating robots between various vibration points. By utilizing a vector method to generate safe flight corridors and optimizing the path with Euclidean signed distance fields, the proposed method reduces the runtime by 80% compared to the original safe flight corridor method and enhances safety by 50%. On embedded systems, the runtime is less than 10 ms. This study is the first to apply the combination of safe flight corridors and Euclidean distance fields for autonomous navigation in concrete vibration tasks, optimizing the original path into a smooth trajectory that stays clear of obstacles. Actual tests of the vibrating robot showed that, compared to traditional methods, the algorithm allows the robot to safely avoid fixed obstacles and moving workers, increasing the execution efficiency of vibration tasks by 60%. Additionally, on-site experiments conducted at three construction sites demonstrated the robustness of the proposed method. The findings of this study advance the automation of concrete vibration work and hold significant implications for the fields of robotics and civil engineering.

## 1. Introduction

When pouring cement during construction, air bubbles may form within, leading to uneven affecting the structural strength of the concrete. Two popular solutions for this issue are as follows: (1) Manual Method: When pouring thin layers of concrete, each layer is compacted by hand. This method is only used for smaller, non-structural concrete applications. (2) Mechanical Method: A common consolidation technique involves using a vibrating rod to vibrate within the cement [1,2], which is suitable for large-scale projects. Currently, the mechanical method primarily relies on manual vibration; however, manual vibration has drawbacks, such as the inability to ensure uniformity, depth, and duration of vibration. Consequently, after a certain period, the building’s walls may develop cracks, leading to structural failures or even collapse [3].

To address the issue of inconsistent manual vibration, it is essential to incorporate intelligent computing technologies, which have seen extensive application in the construction process over the past decade [4]. For instance, semantic methods have been employed to provide formal visual information for construction site images [5], and mobile robots have been utilized to inspect building defects in complex construction environments [6]. To address the issue of low efficiency in manual vibration, this paper proposes an autonomous vibration robot, which replaces manual vibration with a vibrating rod mounted on a robotic arm and is autonomously driven to the vibration points by a tracked chassis. The robot primarily consists of a tracked mobile chassis and a 6-degree-of-freedom robotic arm, which holds the vibrating rod. The chassis autonomously navigates to the preset vibration points, where the vision system determines the vibration position. It then begins the vibration process, and upon confirming that the vibration quality at the current point is acceptable [7,8,9], the vibration ends, and the robot navigates to the next preset point to complete the task. During the autonomous navigation process, the robot needs to plan its path independently. Initially, this article selected the commonly used A* algorithm [10,11] for path planning, but found that this algorithm often resulted in paths too close to obstacles. Since the robotic arm is extended during operation, the robot’s overall center of gravity is high, increasing the risk of collisions or even tipping over, which could damage the robotic arm. To ensure safe and stable operation, the robot needs a path that maintains a safe distance from obstacles. Therefore, this article deployed a safe flight corridor path optimization algorithm [12] that generates paths away from obstacles, addressing the safety issue of collision avoidance in autonomous navigation. However, the safe flight corridor generation has a large range, long computation time, and excessive resource consumption, hindering the robot’s visual recognition capabilities for vibration tasks. To speed up the operation of the safe flight corridor algorithm while still optimizing for safety, this article employed a vector method to generate the corridors, reducing the generation range. This article also applied the Euclidean Signed Distance Field (ESDF) [13] for a comprehensive cost assessment of points within the search range, thereby accelerating the path optimization process. We named this algorithm the ESDF-Vector safe flight corridor (E-VSFC).

## 2. Related Works

### 2.1. The Research and Evaluation Standards Related to Concrete Vibration

Cast-in-place concrete structures, whether piers, columns, or slabs, are among the most common structures in civil and architectural engineering. Introducing vibration during the concrete pouring process ensures proper consolidation of the concrete. Adequate consolidation is crucial for the long-term strength of concrete structures [14]. However, the determination of whether the vibration is sufficient primarily relies on the personal experience of the operator. In many cases, workers operate heavy vibrating equipment, and over time, factors such as fatigue and changes in environmental temperature can lead to inconsistencies in parameters like the location, duration, and depth of vibration. The vibration robot can autonomously navigate to the vibration location with precision, saving time spent on manually transporting the vibration equipment.

### 2.2. Construction Robot Navigation

For the vibration robot to complete its compaction tasks, suitable positioning and navigation algorithms for the cement construction environment are required. A robot developed by Park J. W. uses Ultra-Wideband (UWB) integrated with Building Information Modeling (BIM) [15] technology for indoor construction work [16]. UWB provides high-precision and real-time performance, meeting the positioning needs of indoor construction robots. However, UWB has limited coverage and requires multiple signal base stations, making it unsuitable for the positioning needs of cement vibration robots, and during the cement pouring and vibration stage, the building structure has not yet taken shape, making it impossible to provide usable BIM data. Van Nguyen L. and others developed a robot for automated inspection and assessment in construction [17], equipped with a Global Positioning System (GPS) and an Inertial Measurement Unit (IMU). They used an Extended Kalman Filter (EKF) [18] to fuse the GPS and odometry data for high-precision positioning. However, since cement compaction work may take place in outdoor environments or in closed basement areas, GPS signals can be obstructed. Ren and others used LiDAR, real-time localization technology, and neural networks to generate precise 3D maps [19] has achieved excellent mapping results. However, since the working environment of the vibration robot differs each time, there is no need to construct a high-precision map. Therefore, this article needs a positioning method that is resistant to obstruction, allows for quick deployment, and can be used directly in any scenario. Kim P. et al. [20] employed a SLAM (Simultaneous Localization and Mapping) algorithm that uses 3D LiDAR and infrared sensors to construct a 3D environmental map and provide positioning information. Yang L developed a semantic feature update method based on enhanced visual SLAM [21] for localization and mapping of construction robots.

Building on the approaches of Kim P and Yang L, this article, considering the characteristics of the cement compaction environment and the needs to minimize computational demands, adopts a laser-based SLAM algorithm, selecting the Fast-LIO2 [22] algorithm for localization and obstacle perception. Fast-LIO2 is a more streamlined and lightweight version of Fast-LIO [23] and only requires point cloud data from a 3D LiDAR and pose data from an IMU to provide odometry data at 10-100 Hz, perfectly meeting the positioning needs for cement vibration. For navigation, Kim S. et al. [24] used the ROS-based move base open-source navigation framework for path planning to complete indoor wall spraying tasks. Similarly, Chen X. [25] used the same move base framework to achieve automatic sorting of construction site waste. Karimi S. et al. [26] used the A* algorithm within a custom navigation framework for indoor path planning on the ROS [27] software platform, creating a general construction navigation robot platform. The move base navigation planning section employs Dijkstra’s [28] and A* algorithms [29], which are commonly used path planning algorithms. In summary, the software framework for the navigation component of this construction vibration robot is developed based on ROS and uses the A* algorithm for path planning.

The cement vibration robot needs to complete its vibration tasks while operating safely. However, during actual operation, we have found that the paths planned using the A* algorithm are often too close to obstacles, resulting in frequent collisions that can lead to the robot tipping over and damaging the mechanical arm. During the vibration process, we observed that when the mechanical arm extends from a folded state to start vibrating at each vibration point, this process takes up 60% to 70% of the vibration time, significantly reducing efficiency. Therefore, after the first vibration task is completed, the robot keeps the mechanical arm in an extended position to immediately begin work at the next vibration point. To ensure the robot can safely move with the mechanical arm extended and minimize the number of times it needs to retract the arm to avoid obstacles, this article needs to optimize the A* planned path to be as far from obstacles as possible. For path safety optimization, researchers like Liu have primarily focused on using Dijkstra, A*, or JPS [30] algorithms to generate prior paths, then applying the safe flight corridor [31] (SFC) for safety optimization. Gao et al. [32] generated an initial path in Euclidean Signed Distance Field (ESDF) maps [33,34] and then optimized it using safe flight corridors. Zhou B [35] expanded on Gao’s work by using obstacle gradient information provided by ESDF for path safety optimization. Zheng proposed an analytical construction method for density functions to address navigation problems with safety constraints almost everywhere [36], specifically for safe planning of robots. When passing near obstacles, the robot autonomously moves away from them. The handling of obstacles is similar to the ESDF. Lee et al. proposed a path planning method for quadruped robots capable of traversing complex terrains, considering terrain traversability [37]. This study mainly focuses on robot path planning in the rebar vibration environment, where the rebar mesh is relatively flat without complex changes. However, it provides valuable insights for the research of other construction robots in the future. Building on these studies, this paper applies the safe flight corridor algorithm to optimize the path, enabling the vibration robot to navigate safely. However, the long computation time and high resource consumption prevent the robot’s visual system from completing the vibration task. Therefore, this paper combines the method with ESDF. First, this article reduces the extension range of the safe flight corridor using a vector method. Then, to improve optimization efficiency within the corridor, this article uses ESDF for comprehensive cost assessment to quickly select optimized path points, achieving rapid and safe path optimization.

## 3. Materials and Methods

### 3.1. Research Framework

The hardware platform of this paper consists of a cement vibration system, which is a combination of a tracked walking system and a mechanical arm grasping system, as shown in Figure 1. In building the hardware system, we referred to Timo’s hardware design framework for building inspection robot systems [6], and the sensor part of the tracked walking system is composed of a Livox 3D LiDAR(RoboSense, Shenzhen, Guangdong, China) and a high-precision IMU(Wheeltec Technology Co., Ltd, Guangzhou, Guangdong, China). The upper-level control system uses an Nvidia Jetson Xavier controller(NVIDIA Corporation, Santa Clara, CA, USA), which is equipped with an Ubuntu 20.04(Canonical Ltd, London, United Kingdom) embedded system. The lower-level control system consists of an STM32F407IGH6(STMicroelectronics, Geneva, Switzerland) MCU and a servo controller for brushless DC motors. Both the localization and navigation algorithms are deployed on the Nvidia controller. The upper-level system sends linear and angular velocity commands to the STM32 via a USB-to-CAN module, which then calculates and uses a PID controller to control the robot’s movement.

This study designed a GUI interface for controlling the robot. Various functions of the GUI interface are shown in Figure 2. Through this GUI interface, the working mode of the robot can be set, and the vibration condition, position, and attitude data of the robot can be monitored in real time. The vibration depth can also be set, and in case of unexpected situations, instructions can be issued to manually stop the robot. Additionally, the CAD map of the construction site can be displayed. As shown in Figure 3, the vibration robot can automatically navigate to each vibration point by simply marking the vibration locations on a CAD drawing and dragging them onto the robot’s monitoring interface. At the starting point, the robot arm is in a retracted state (Figure 3a). During movement, the robot arm remains in a folded state (Figure 3b). As the robot approaches the vibration point, the robot arm extends into a preparation state (Figure 3c), and upon reaching the vibration point, the robot arm begins the vibration task (Figure 3d). While moving to the next vibration point, the robot arm remains in an extended state (Figure 3e), and upon reaching the next point, it starts vibrating again (Figure 3f). After completing all vibration tasks, the robot returns, folding the robot arm during the return journey (Figure 3g), and ultimately returns to the starting point, completing the vibration work (Figure 3h).

During actual operation, this article found that the robot arm moves from a folded state to an extended state at each vibration point and then returns to a folded state afterward, which takes up to 60% of the vibration time. Therefore, throughout the entire process from the initial vibration task to the final return, the robot arm should remain in an extended posture. Consequently, the navigation system must ensure that the robot can move safely with the robot arm extended, and it should only retract the robot arm when encountering very narrow spaces where the robot can only pass in the folded state. During the actual operation, we found that the arm moves from a folded state to an extended state at each vibration point and then returns to a folded state afterward, which takes up to 60% of the vibration time. Therefore, throughout the entire process, from the initial vibration task to the final return, the arm should remain in an extended posture. Consequently, the navigation system must ensure that the robot can move safely with the arm extended, and it should only retract the robot arm when encountering very narrow spaces where the robot can only pass in the folded state.

To address the safety and efficiency requirements of the robot arm, this article established a navigation framework for the construction robot. This article referenced the ROS built-in move base [38,39] navigation open-source algorithm framework. However, the move base framework does not consider the safety of the robot arm, and each time before vibration tasks, it takes over half an hour to construct a map [40]. Additionally, since the construction sites vary, the constructed maps are not universally applicable. Therefore, this article developed a navigation software system for construction robots based on ROS, as shown in Figure 3. This article employed the Fast-LIO2 [41] algorithm for real-time mapping of the work scene, using the odometry information from Fast-LIO2 for robot localization. The point cloud data (PCD) generated by Fast-LIO2 is first processed by layering and then merging to produce a 2D map required for robot navigation. Subsequently, this article performed a dilation operation on the 2D map, using the maximum ground projection size of the robot chassis as the dilation radius. After generating the dilated map, we constructed the ESDF map.

Based on the previous analysis of the concrete vibration robot’s workflow in the Figure 3, this article needs a path optimization algorithm that generates routes as far away from obstacles as possible, ensuring the robot arm can safely pass through obstacles while in an extended state. Initially, this article used the safe flight corridor algorithm for path optimization, which generated safe paths. Although the optimized paths met safety requirements and allowed the robot arm to pass safely through obstacles, this article discovered that the optimization process took a long time and consumed significant computational resources, hindering the robot’s visual system from completing the vibration task. Through extensive experimental testing, this article found that the slow optimization of path points and the long optimization time were due to the large corridor expansion range in the safe flight corridor algorithm, which resulted in a high time complexity for selecting path optimization points; the path planning framework of the vibration robot is shown in Figure 4.

### 3.2. Generation of Safe Flight Corridor Based on Vector Method

The safe flight corridor is expanded using rectangles, circles, or other convex polygons. Optimization points are selected within the corridor to generate the optimized path. In Figure 5, the green line represents the original path, and the blue line is the optimized path. Initially, the construction robot uses the safe flight corridor as shown in Figure 5a, where the yellow transparent area is the expanded corridor, and the blue line is the optimized path. However, for the construction robot, the red transparent area in Figure 5b is sufficient for generating an optimized path, and the yellow transparent area includes much unnecessary redundancy compared to the red area.

How to generate the red transparent area of the safe flight corridor is the problem this article needs to solve. As can be seen, the red area corridor is generated based on the path’s direction and trend. If we consider the path as a function curve of y with respect to x on a 2D map, this article could use the derivative method to calculate the tangent slope at each path point, and then use the tangent to find the normal, thereby generating the safe flight corridor.

#### 3.2.1. Solving Path Normals Based on Vector Calculation

This article verified the method of using derivatives to calculate the tangent slope at path points, and then constructing the normal line based on the tangent slope. Considering that calculating derivatives requires at least two points, fitting a curve with more points leads to a more accurate solution for the tangent slope. However, using more points increases the computational complexity and time required. Therefore, this article used the method of calculating the tangent slope at the midpoint of three adjacent points to determine the tangent at each path point. This article selects three adjacent points, A, B, and C, and fits a curve, as shown by the blue curve in Figure 6a. The curve function at this point is a quadratic polynomial of y with respect to x, as shown in Equation (Equation 1). Before solving for the tangent slope at point B, this article needsto solve for $\left(a_{0},a_{1},a_{2}\right)$. Let **A** = ${\left(a_{0},a_{1},a_{2}\right)}^{T}$, **Y** = ${\left(y_{0},y_{1},y_{2}\right)}^{T}$, and X be as shown in Equation (Equation 2). This article constructs Equation (Equation 3) to solve for A. Once A is solved, as shown in Equation (Equation 4), we can calculate the tangent slope $k_{b}$ at point B. ***N*** is the normal vector at point B, and $k_{N}$ is the slope of the normal vector. After solving for $k_{b}$, we can solve for $k_{N}$. The downside of this method is that when the path is horizontal or vertical, as shown by the blue line in Figure 6b, $k_{b}$ will encounter the special case shown in Figure 6b, where it becomes impossible to solve for the slope of the normal vector $k_{N}$.(1)$$y=a_{0}+a_{1}*x+a_{2}*x^{2}$$(2)$$X=\left[\begin{array}{ccc} 1 & x_{1} & x_{1}^{2}\\ 1 & x_{2} & x_{2}^{2}\\ 1 & x_{3} & x_{3}^{2} \end{array}\right]$$(3)$$\left\{\begin{array}{c} Y=XA\\ A=X^{-1}Y \end{array}\right.$$(4)$$\left\{\begin{array}{c} k_{b}=a_{1}+2a_{2}x_{2}\\ k_{N}=\frac{1}{k_{b}} \end{array}\right.$$

Since there are cases in Figure 6b where the normal vector cannot be solved, a method is needed that can solve for the normal vector in any path condition or an approximate solution method. Based on the relationship between the three path points, this article attempts to use a vector-based approach, as shown in Figure 7a. Since the path points generated by the front-end planning algorithm are evenly spaced, this article assumes that every three points A, B, and C form an isosceles triangle. Due to the isosceles triangle, the slope $k_{ac}$ is approximately equal to $k_{b}$, as shown in Figure 7b. By solving for the midpoint $M_{AC}$ of the base AC, the line from vertex B to midpoint $M_{AC}$ is perpendicular to the base AC. Therefore, this article considers the line ${\mathrm{BM}}_{\mathrm{AC}}$ to be the normal vector at midpoint B, as shown in Figure 7a. The solution process is detailed in Equation (Equation 5) where the function unit calculates the unit vector of ${\mathrm{BM}}_{\mathrm{AC}}$. During the sampling process, this article samples the mapping of the original path points onto the grid coordinate system to avoid repeated sampling. The three points A, B, and C ensure that they are non-repeating and adjacent in the grid coordinate system.(5)$$\left\{\begin{array}{c} A=\left(x_{1},y_{1}\right)\;B=\left(x_{2},y_{2}\right)\;C=\left(x_{3},y_{3}\right)\\ M_{AC}=\left(A+B\right)/2\\ {\mathrm{BM}}_{\mathrm{AC}}=M_{AC}-B\\ Unit\left({\mathrm{BM}}_{\mathrm{AC}}\right)={\mathrm{BM}}_{\mathrm{AC}}/∥{\mathrm{BM}}_{\mathrm{AC}}∥ \end{array}\right.$$

The vector method assumes an ideal case based on an isosceles triangle and uses an approximation to solve for the normal vector, providing an approximate value of the true normal vector. Therefore, it is necessary to perform a significance test on the differences between the slopes of the path points calculated by the derivative method and the vector method to verify the effectiveness of the vector method. A total of 100 random paths were sampled and analyzed, and as shown in Figure 8, there is no significant difference between the two methods in calculating the slopes of the path points, indicating that the vector method is valid. A random selection of 10 sets of path samples is shown in Table 1. The time used by the vector method is 1.56% of the time used by the derivative method, demonstrating that the vector method has a much faster computation speed.

#### 3.2.2. Constructing a Safe Flight Corridor

Based on the unit vector ${\mathrm{BM}}_{\mathrm{AC}}$, the corridor is expanded by a set length. In Figure 9, the set length is 5 points (5 points on one side, 10 points on both sides; the reason for using five points on one side is to conveniently demonstrate the algorithm principle in the figure), resulting in the green safe flight corridor shown. The light green area in Figure 9 indicates the region that is expanded once, while the dark green area represents the overlapping expansion region, the purple dot represents the starting point, the red dot represents the midpoint, the orange represents the original path, and the black areas represent obstacles. However, in Figure 9, the corridor section marked by the blue circle is interrupted, which can pose a risk of the optimized path being too close to obstacles, failing to meet safety requirements. To address this issue, after obtaining the A* path, this article performs a pre-expansion based on the vector relationships of the A* path points to explore the potential corridor area that can be generated. The boundary of the pre-expansion, indicated by the blue lines in Figure 10, defines the limits of the corridor that can be created. Once the boundaries are determined, this article generates the safe flight corridor, as shown in Figure 10. The light green area indicates regions that are expanded once, while the blue area represents overlapping expansion. At this point, selecting optimization points within the corridor to generate the optimized path is safe, effectively resolving the risks present in Figure 9.

### 3.3. Conduct Comprehensive Cost Evaluation in ESDF

The corridor can achieve the goal of safe hard constraints, but there may still be situations where it is close to obstacles. To ensure the safety of the vibration robot, the algorithm incorporates the ESDF as an additional safety constraint. At the same time, the ESDF speeds up the selection of path points within the corridor, saving runtime. The method involves filtering path points within the corridor based on the magnitude of the ESDF values, prioritizing the selection of points with smaller ESDF values as optimization paths, as shown in Figure 11. The red point is the endpoint, yellow represents the original path, the black areas represent obstacles light blue indicates the points on the original path at turning points, and dark blue shows the corridor generated at the turning points. The purple points are the optimized path points, and the orange line with arrows represents the optimized path. Although the generated optimized path is indeed safe, sharp turns appear in the red circle area of Figure 11, making it difficult for the robot to execute the path and increasing the risk of tipping over.

To address the issue of sharp turns in the optimized path, the algorithm adds the distance cost $C_{end}$ from the corridor point $P_{now}$ to the endpoint $P_{end}$ on top of the safety cost $C_{esdf}$. It also includes the distance cost $C_{last}$ from the corridor point $P_{now}$ to the previous point $P_{last}$ and the repeat exploration cost $C_{repeat}$. As shown in Equation (Equation 6), the original path point R in the corridor serves as a reference, where $R_{last}$ represents the distance from the original path point R to the previous point $P_{last}$, and $R_{end}$ represents the distance from the original path point *R* to the endpoint $P_{end}$. The potential field value of the original path point *R* compared to the maximum potential field value serves as the risk assessment coefficient $R_{risk}$, while the distance $Dis_{max}$ between $R_{end}$ and the original path is used as the energy consumption coefficient $R_{energy}$. By introducing $R_{last}$,$R_{end}$,$R_{risk}$,$R_{energy}$, the algorithm resolves the issue of sharp turns in the path shown in Figure 11, generating an optimized path, as depicted in Figure 12. This ensures that the robot can travel safely without tipping over due to sharp turns in the path. In $C_{last}$, $R_{last}$+1 prevents the denominator from becoming zero, and similarly, $R_{end}$+1 in $C_{end}$ also prevents the denominator from being zero. To make the algorithm adaptable to different scenarios, coefficients $k_{esdf}$,$k_{repeat}$,$k_{last}$,$k_{end}$ have been added; in Equation (Equation 6), K is ${\left(k_{esdf},k_{repeat},k_{last},k_{end}\right)}^{T}$ and C is ${\left(C_{esdf},C_{repeat},C_{last},C_{end}\right)}^{T}$, allowing the parameter values to be set according to various application scenarios and requirements to generate the desired optimized path. Algorithm  1 for the E-VSFC algorithm is as follows.
**Algorithm 1:** E-VSFC Path Optimization**Data**:originalPath**Result**:optimizedPath// Use vector method to calculate the normal vector of the original path points**for** 
i←1tolength(originalPath)do    pathPoint←originalPath[i]    pointVector←Calculate_path_point_Vector(pathPoint)    normalVector.append(Calculate_the_Normal_vector(pointVector))// Explore boundaries and generate SFC**for** 
i←1tolength(normalVector)do    boundary←Pre_expansion(normalVector[i])    extendedCorridorPart←extendCorridor(boundary)    SFC.append(extendedCorridorPart)// Solve the optimized path in ESDF using comprehensive cost.**while**(pathPoint!=endPoint)    OptimizedPoint←ComprehensiveCost(SFC)    optimizedPath.append(OptimizedPoint)**return** 
optimizedPath(6)$$\left\{\begin{array}{c} R=\left(x_{1},y_{1}\right)\;P_{now}=\left(x,y\right)\\ P_{last}=\left(x_{last},y_{last}\right)\;P_{end}=\left(x_{end},y_{end}\right)\\ Dis_{max}=\\ \;\sqrt{{\left(x_{start}-x_{end}\right)}^{2}+{\left(y_{start}-y_{end}\right)}^{2}}\\ R_{last}=\\ \;\sqrt{{\left(x_{1}-x_{last}\right)}^{2}+{\left(y_{1}-y_{last}\right)}^{2}}\\ R_{end}=\\ \;\sqrt{{\left(x_{1}-x_{end}\right)}^{2}+{\left(y_{1}-y_{end}\right)}^{2}}\\ R_{risk}=ESDF\left(P_{now}\right)/ESDF_{max}\\ R_{energy}=R_{end}/Dis_{max}\\ C_{esdf}=\frac{ESDF\left(P_{now}\right)*R_{energy}}{ESDF_{max}}\\ C_{last}=\\ \;\frac{\sqrt{{\left(x-x_{last}\right)}^{2}+{\left(y-y_{last}\right)}^{2}}}{\left(R_{last}+1\right)}*\left(1-R_{risk}\right)\\ C_{end}=\\ \;\frac{\sqrt{{\left(x-x_{end}\right)}^{2}+{\left(y-y_{end}\right)}^{2}}}{\left(R_{end}+1\right)}*\left(1-R_{risk}\right)\\ Cost=K^{T}*C \end{array}\right.$$

## 4. Results and Discussion

### 4.1. Experimental Configuration

In our experiment, the simulation platform was configured with Ubuntu 20.04 OS, 32.0 GB RAM, Core i7-10700K 2.6 GHz CPU, and P620-dedicated GPU. This article conducted safety comparison tests between the E-VSFC-optimized path and the original path, as well as performance comparison tests between E-VSFC, SFC, and A*-ESDF in terms of runtime, number of expanded nodes, and optimized path length. The construction robot was equipped with a Jetson Xavier NX computing platform, a mid360 LiDAR, a mobile chassis, and a six-degree-of-freedom industrial robotic arm, as shown in Figure 1.

### 4.2. Safety Comparison Between Optimized Path and Original Path

The primary goal of optimizing the navigation path for construction robots using the E-VSFC algorithm is safety. The calculation method of the Euclidean Signed Distance Field is as shown in Equations (Equation 7) and (Equation 8); in the ESDF, the distance field computation process involves calculating the minimum distance from each free voxel **p** in the map to the nearest obstacle voxel **q**; the smaller the minimum distance, the closer it is to the obstacle and the danger it is. This article used the average Euclidean Signed Distance Field (ESDF) value of the path points as a safety measure; the smaller the average ESDF value of the path points, the safer the path is considered. Before applying the algorithm in practice, safety verification is required. Our verification platform is a simulation system based on the ROS platform. This article randomly generated 10 maps. Figure 13a shows one of the generated maps, and Figure 13b shows the ESDF processing for that map. Randomly generated maps have dimensions greater than 900 $m^{2}$, with a grid resolution of 0.01 $m^{2}$.(7)$$SDF\left(p\right)=\left\{\begin{array}{cc} +\min_{q\in O}{∥p-q∥}_{2}, & p\in F\;(\mathrm{Free}\;\mathrm{space})\\ -\min_{q\in F}{∥p-q∥}_{2}, & p\in O\;(\mathrm{Obstacle}) \end{array}\right.$$(8)$${∥p-q∥}_{2}=\sqrt{{\left(x_{p}-x_{q}\right)}^{2}+{\left(y_{p}-y_{q}\right)}^{2}}$$

In 10 randomly generated maps, a total of 100 experiments were conducted per map, resulting in 1000 experiments. In each experiment, the target point and endpoint were randomly selected on the map. This article performed a significance analysis on the final data. Table 2 shows 10 randomly selected experiment results, summarizing the total number of path points, the total ESDF values, and the average ESDF values. ESDF values were used as the safety evaluation standard, with lower values indicating safer paths. The average ESDF value of the E-VSFC-optimized paths was 23% lower than that of the A* paths, and the path points optimized by the E-VSFC have a greater distance to obstacles and therefore higher safety compared to the path points generated by the A* algorithm. Figure 14 shows the significance analysis of the differences in average ESDF values between E-VSFC-optimized paths and original paths from 100 random trials conducted on 10 maps. The green line represents the average ESDF value within the obstacle-free range of the map, which is used as a criterion to evaluate the obstacle density of the map. The symbol “***” indicates that the p-value is much less than 0.001; the statistically significant average ESDF value of the optimized path points using E-VSFC is lower than the average ESDF value of the map. This demonstrates that the E-VSFC algorithm is effective in safely optimizing the original paths.The presence of outliers in E-VSFC and A* in Figure 14 is due to the random selection of start and end points in areas with very dense obstacles, as shown in Figure 15: black is obstacles, purple is start point, red is end point, and green is generated corridor. In such cases, the constructed corridor width is narrow, resulting in less noticeable optimization effects. Consequently, under these circumstances, the results of E-VSFC and A* may be comparable. Figure 16 shows the optimized paths in the map from Figure 13. The blue path is A*, while the red path is E-VSFC-optimized. In the green-circled areas, the E-VSFC-optimized path maintains a greater distance from obstacles compared to the A* path. In conclusion, E-VSFC effectively enhances path safety optimization.

### 4.3. Performance Comparison Testing of the E-VSFC Algorithm

#### 4.3.1. Algorithm Simulation Experiment

The tracked chassis is a common differential-drive chassis. It offers more stable operation compared to wheeled chassis, and its pressure distribution is more uniform, which prevents damage to the bundled rebar structure. As shown in the figure, this is the 3D diagram of the tracked chassis, where ***L*** represents the length of the tracked chassis, and ***w*** represents the width of the tracked chassis. The distance between the centers of the two tracks is $w_{mid}$. Maximum mass of the vibration robot is $m_{max}$. The kinematic model of the tracked chassis is shown in Equation (Equation 9), *v* is the linear velocity of the chassis, ω is the angular velocity of the chassis, $R_{t}$ is the turning radius, *x* and *y* are the displacements of the chassis, and θ is the yaw angle of the chassis. The main geometric and motion parameters of the chassis are shown in Table 3.

This article conducted path optimization experiments with random starting and ending points for E-VSFC, safe flight corridors, and A* + ESDF on the 10 randomly generated maps shown in Figure 13. Each map was run 100 times, totaling 1000 experiments. This article selected runtime, the number of algorithm expansion nodes, and path length as performance evaluation criteria. The algorithm parameter settings are shown in Table 4. The parameters in Table 4 are set according to the parameters of the tracked chassis (see Table 3: Main Parameters of the Chassis). According to Equation (Equation 9), the minimum value of the robot’s **$R_{t}$** is 0.2 m. In the actual vibration environment, the maximum allowable speed of the robot was tested to be 0.5 m/s (exceeding this speed would damage the low-grade steel bar bundle structure). According to Equation (Equation 11), the normal acceleration **$a_{n}$** is calculated as 1.25 m/s^2^. From a dynamics perspective, the robot can stably operate with *R_t_* at 0.2 m. Since the maximum *R_t_* for a differential-drive chassis is theoretically infinite, the allowable path curvature is therefore very high. The parameters of the E-VSFC algorithm are set according to the principle of prioritizing safety. The corridor width is set to 2.0 m, which is larger than the chassis length listed in Table (Main Parameters of the Chassis). The trajectory-following accuracy of the chassis can achieve a positional accuracy of 0.2 m, and the attitude-following accuracy (primarily the yaw angle) can reach π/18, as shown in Figure 17; *e_d_* represents position following error, *e_θ_* represents attitude angle error, ***w*** is the width, and ***L*** is the length. The black line indicates the optimized path, while the green area represents the safe flight corridor. Based on Equation (Equation 10), calculated $error_{max}$ is 0.553 m, which is much smaller than half the width of the corridor, ensuring safety. Therefore, a corridor width of 2.0 m fully satisfies the safety requirements. The remaining coefficients were tested in the simulation experiments and demonstrated the best performance. The coefficients for the safe flight corridor are directly set according to the actual performance parameters of the tracked chassis Table 3, and the parameter settings for the A*-ESDF algorithm are the same as those described in Section 4.2.(9)$$\left\{\begin{array}{c} v=(v_{l}+v_{r})/2\\ \omega =\left(v_{r}-v_{l}\right)/w_{mid}\\ R_{t}=w_{mid}\cdot \left(v_{r}+v_{l}\right)/\left(2v_{r}-2v_{l}\right)\\ \dot{x}=v\cdot cos\left(\theta \right)\\ \dot{y}=v\cdot sin\left(\theta \right)\\ \dot{\theta }=\omega \end{array}\right.$$(10)$$error_{max}=e_{d}/*cos\left(\theta \right)+w/2$$(11)$$v=v^{2}/R_{t}$$

This article conducted path optimization experiments with random starting and ending points for E-VSFC, SFC, and A*-ESDF on the 10 randomly generated maps shown in Figure 13. Each map was run 100 times, resulting in a total of 1000 experiments. This article selected runtime, the number of algorithm expansion nodes, and path length as performance evaluation criteria, with algorithm parameter settings shown in Table 4. The comparison of algorithm performance parameters is illustrated in Figure 18, where Figure 18a shows the comparison of algorithm runtime and Figure 18b shows the comparison of the number of expanded nodes (*** indicates a p-value far less than 0.001, statistically significant). E-VSFC achieved the fastest runtime at 7.9% of SFC’s time and 39.6% of A*-ESDF’s time, while it had the fewest accessed nodes at 11.2% of SFC’s count and 26.9% of A*-ESDF’s count. Figure 18c shows that there are no significant differences in path lengths among the three algorithms, indicating that E-VSFC has similar energy consumption to SFC and A*-ESDF. Figure 19 illustrates the effect of optimized paths in the random maps. Overall, compared to SFC and A*-ESDF, E-VSFC has the shortest runtime and fewest expanded nodes while maintaining a similar path length, suggesting that E-VSFC performs the best.

Based on the validity verification and simulation test results, E-VSFC has achieved path safety optimization and demonstrates faster runtime and fewer expanded nodes compared to SFC, occupying less computational resources. Ten randomly selected sets of experimental data are shown in Table 5: the data with the shortest runtime, shortest path, and fewest expanded nodes are highlighted in bold. The effectiveness of E-VSFC and SFC in generating safe flight corridors is illustrated in Figure 20, where the red transparent area represents the corridor generated by E-VSFC, which is smaller in scope than the yellow transparent area representing the corridor generated by SFC. In summary, E-VSFC has achieved our goal of rapid path safety optimization with a faster runtime, fewer expanded nodes, and the construction of smaller safe flight corridors while consuming fewer computational resources.

#### 4.3.2. Algorithm Real-World Execution Experiment

With the permission of the construction company, the robot conducted 100 runs for algorithm testing in the actual construction environment shown in Figure 21. Figure 21a is a CAD drawing; due to the construction drawings, the construction company only allows the overall dimensions to be marked in this paper, and detailed dimensions cannot be disclosed. Figure 21b is the map established by SLAM for algorithm testing, the hardware uses the Mid360 LiDAR and BMI088 sensor, while the software employs the Fast-lio2 algorithm. We process the collected PCD data by layering it according to the method shown in Figure 4 and then merge it to construct the map. The actual construction environment is 81 m length, 33 m width, overall area of approximately 2000 $m^{2}$, and SLAM mapping resolution is 0.01 $m^{2}$.

The results of the significance analysis of the actual operational differences are shown in Figure 22, while Figure 23 presents the effect of optimized paths in the actual construction environment. E-VSFC achieved the fastest runtime, taking 11% of SFC’s time and 25% of A*-ESDF’s time. It also had the fewest accessed nodes, with 12.3% of SFC’s count and 17.3% of A*-ESDF’s count. The differences in algorithm performance results are consistent with those observed in simulation conditions. E-VSFC met the expected objectives, showing significantly shorter runtime than SFC in actual engineering operations, along with fewer expanded nodes and lower computational resource usage. Ten randomly selected sets of experimental data are shown in Table 6.

Table 7 includes the number of static obstacles (such as construction equipment, rebar, etc.) and moving obstacles (such as workers, cement pouring machines, etc.). The number of obstacles is calculated based on the number of static obstacles the robot needs to avoid, and the sudden appearance of workers during its journey from the starting point to the endpoint, the number of static obstacles does not include the inherent obstacles of the construction site. All static obstacles in the table are construction work equipment present during the test. The execution results of the A* algorithm and the E-VSFC algorithm include the stopping radius, whether the vibration task was completed, robot runtime, and whether any collisions occurred during operation. From the data in the table, it can be seen that the E-VSFC algorithm plays a role in safe obstacle avoidance, ensuring no collisions occur during the robot’s operation.

Figure 24 shows the construction robot deploying the E-VSFC algorithm to perform vibration tasks on the construction site. In Figure 24a, the vibration robot starts from the initial point and moves toward the target vibration position in Figure 24l. From Figure 24b–e, E-VSFC constructs a safe flight corridor between workers and vertical steel bars in real time to optimize the path, allowing the vibration robot to pass safely. In Figure 24g–i, the vibration robot avoids obstacles by following the re-optimized path. In Figure 24j–l, when workers (in green in the figure) randomly move near the vibration robot, E-VSFC still optimizes a safe path, ensuring smooth passage. The experiment demonstrates that the E-VSFC algorithm can optimize safe paths in both static and dynamic environments, ensuring the smooth passage of the vibration robot. E-VSFC consumes fewer computational resources, allowing the embedded platform to allocate more resources to the robot’s vision system, resulting in better cement vibration effects. Consequently, the vibration robot successfully completed the vibration task with improved performance.

### 4.4. Conclusions

The contribution of this paper is to provide a path safety and obstacle avoidance navigation method for a vibrating robot based on a tracked chassis, which improves the efficiency of path planning during the vibration process and reduces the time required for automatic vibration. Relying solely on paths generated by A* does not meet the safety requirements for robot operation; the robotic arm frequently folds and extends during the vibration process, occupying 60–70% of the working time. While SFC meets safety needs, it has long computation times and consumes significant resources, negatively impacting the recognition performance of the robot’s vision system and preventing it from completing the vibration task. To ensure the robot can operate safely and complete the vibration task, this study proposes a path optimization algorithm for automated navigation of the vibration robot. It employs the Euclidean Signed Distance Field and safety flight corridors generated by a vector-based method (E-VSFC) to optimize the path safely.

To ensure the reliability of the algorithm, multiple maps were randomly constructed for simulation testing. It was confirmed that the E-VSFC-optimized paths are sufficiently safe, requiring fewer expanded nodes compared to SFC, with a smaller range for the constructed safety flight corridors, faster runtime, and less computational resource usage. When the E-VSFC algorithm was applied in real scenarios, the actual operational performance matched that of the simulations, achieving the goal of safe and rapid path optimization with low resource consumption. These advancements will promote the development and application of intelligent construction techniques, enhancing the quality of concrete reinforcement. The algorithm has already been tested in a development project at a computing center, and future work will extend to more construction engineering applications. In the future development of the algorithm, the replanning method FRT* designed by Li et al. [42] can be considered. The replanning idea can be applied to the vibration robot to enhance its environmental adaptability.

## Figures and Tables

**Figure 1 sensors-25-01765-f001:**
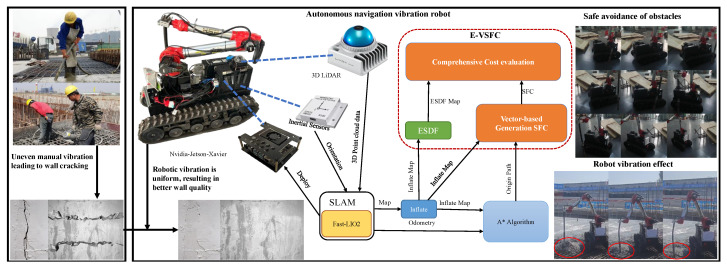
Diagram illustrating the hardware system framework of the vibration robot. The red circle highlights the research content of this study.

**Figure 2 sensors-25-01765-f002:**
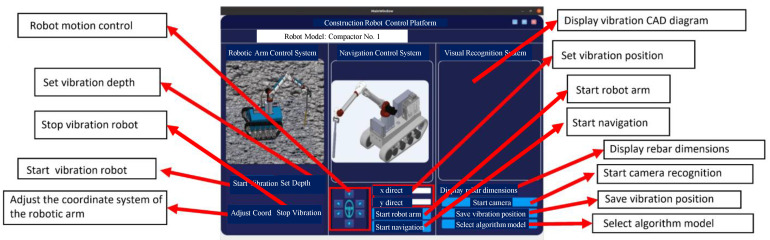
Explanation of the graphical user interface (GUI) functions for the vibration robot.

**Figure 3 sensors-25-01765-f003:**
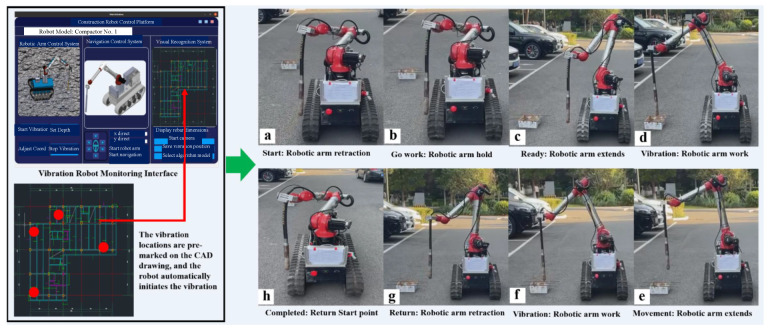
Process flow diagram of the vibration robot system.

**Figure 4 sensors-25-01765-f004:**
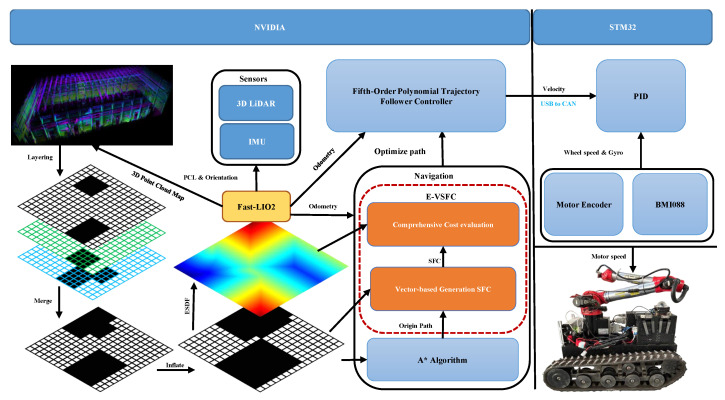
Research framework diagram of the vibration robot software system.

**Figure 5 sensors-25-01765-f005:**
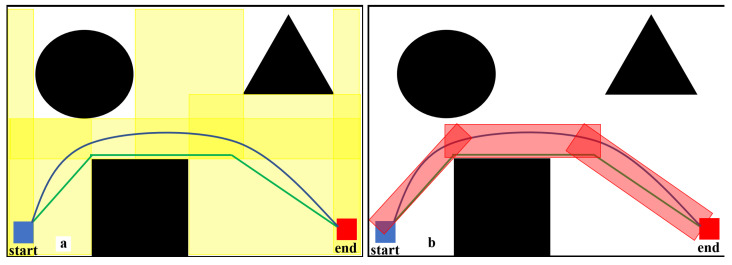
Safe flight corridor generation diagram. The black sections in the figure represent obstacles. (**a**) shows the corridor range generated by the traditional method for optimizing the path. (**b**) shows the corridor range required for the actual optimized path.

**Figure 6 sensors-25-01765-f006:**
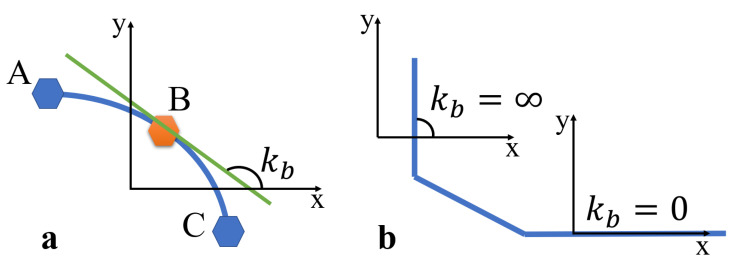
Derivative method. (**a**) Under normal conditions, the slope at point B is solved by fitting a curve through points A, B, and C. (**b**) When points A, B, and C are aligned in a vertical or horizontal line, solving for the slope at point B results in no solution.

**Figure 7 sensors-25-01765-f007:**
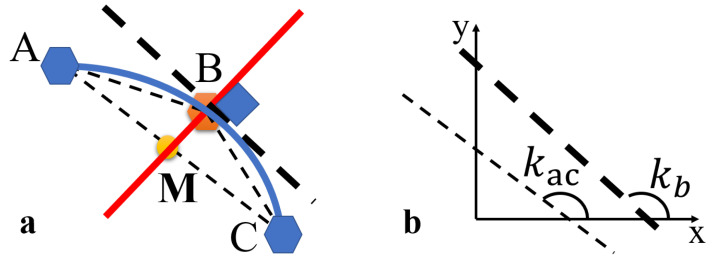
Vector method. (**a**) The vector method is used to solve for the normal vector at point B. (**b**) The difference between solving at point B using the vector method and the conventional method.

**Figure 8 sensors-25-01765-f008:**
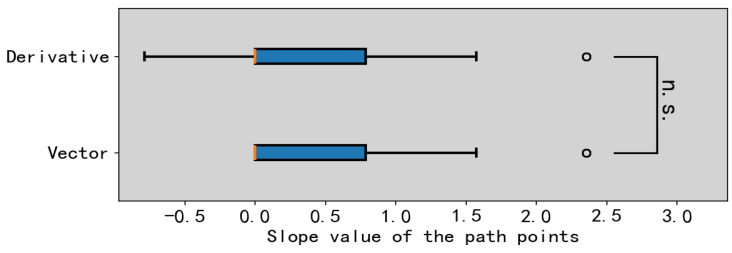
Data comparison of differences between vector method and derivative method for solving the normal vector slope.

**Figure 9 sensors-25-01765-f009:**
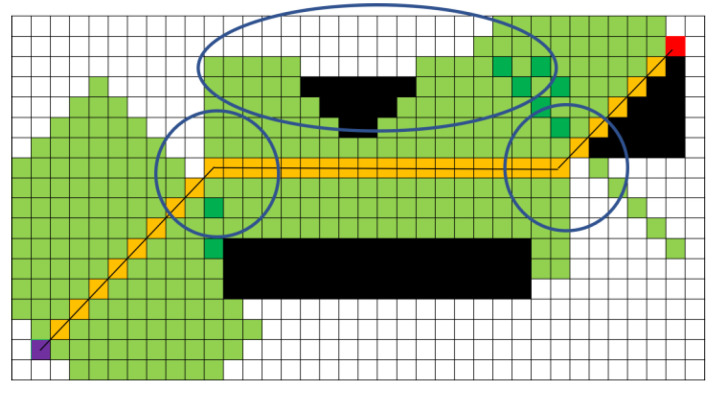
The corridor generated without pre-expansion.

**Figure 10 sensors-25-01765-f010:**
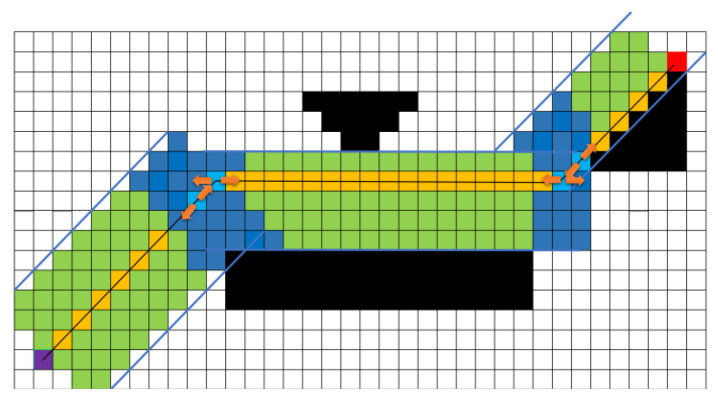
The corridor generated with pre-expansion.

**Figure 11 sensors-25-01765-f011:**
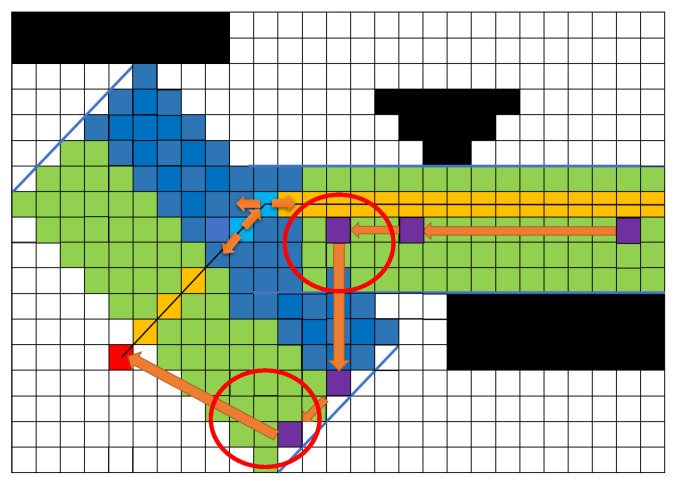
Only consider the path optimized by ESDF.

**Figure 12 sensors-25-01765-f012:**
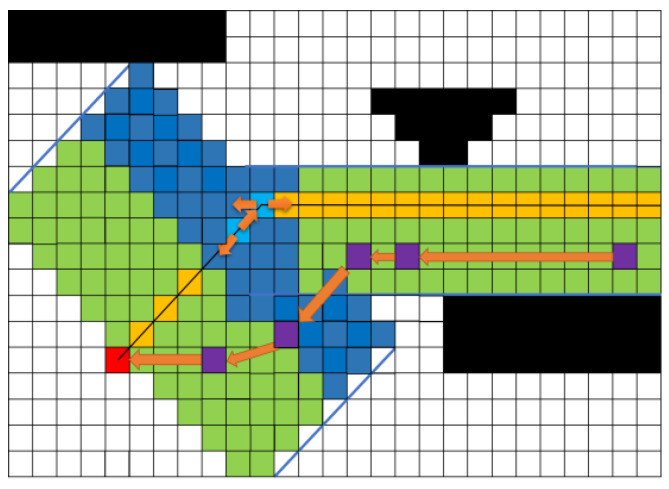
Comprehensive cost assessment.

**Figure 13 sensors-25-01765-f013:**
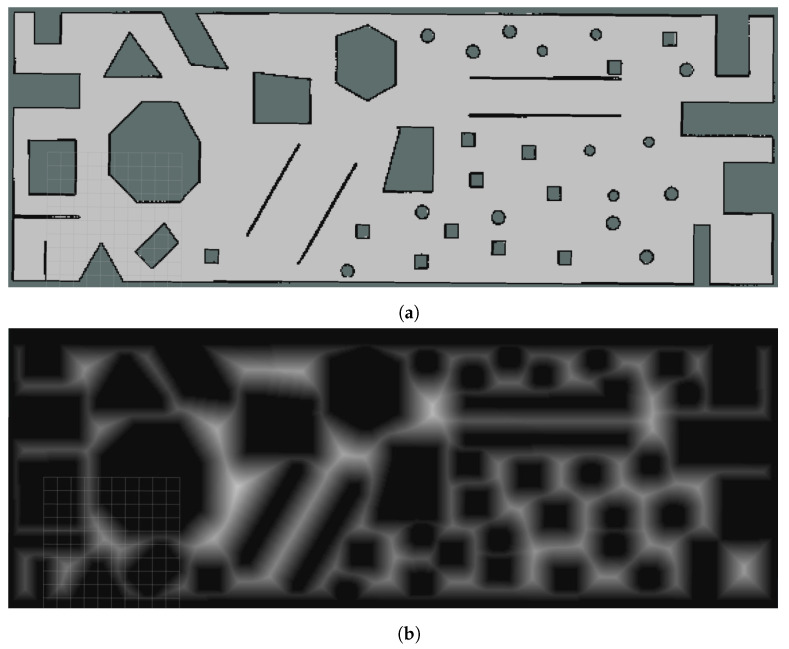
Randomly generated and ESDF processing maps. (**a**) Randomly generated maps. (**b**) ESDF processing of randomly generated maps.

**Figure 14 sensors-25-01765-f014:**
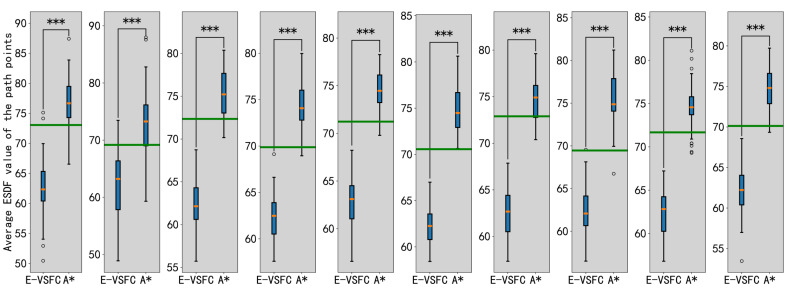
Significance analysis of the difference in ESDF values between E-VSFC-optimized path and A*-generated paths.

**Figure 15 sensors-25-01765-f015:**
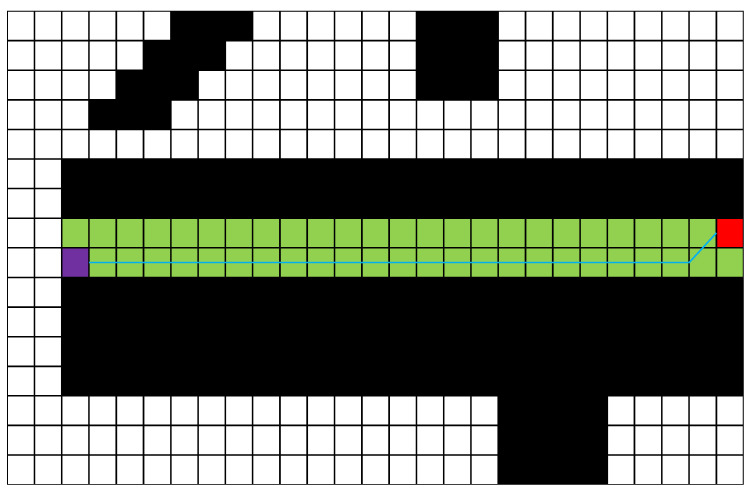
The situation where the initial path generates a very narrow corridor in densely obstructed areas, resulting in minimal optimization effects.

**Figure 16 sensors-25-01765-f016:**
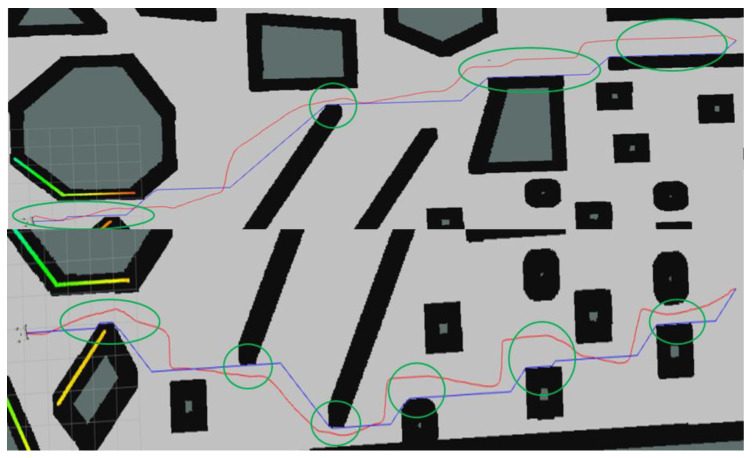
Comparison of E-VSFC-optimized paths and A*-generated paths.

**Figure 17 sensors-25-01765-f017:**
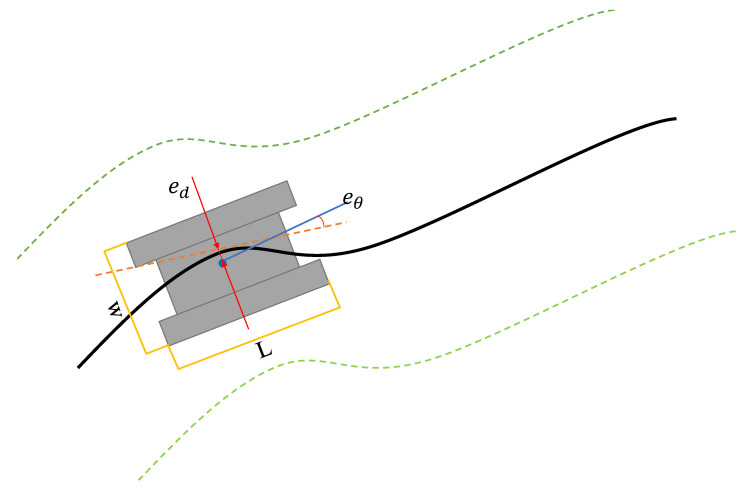
Tracked chassis for vibration robots.

**Figure 18 sensors-25-01765-f018:**
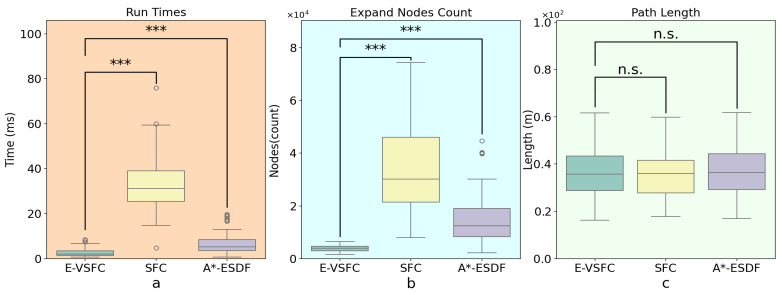
Comparison of path optimization results for E-VSFC, SFC, and A*-ESDF on simulation maps. (**a**) Comparison run times (**b**) Comparison number of expanded nodes (**c**) Comparison optimized path lengths.

**Figure 19 sensors-25-01765-f019:**
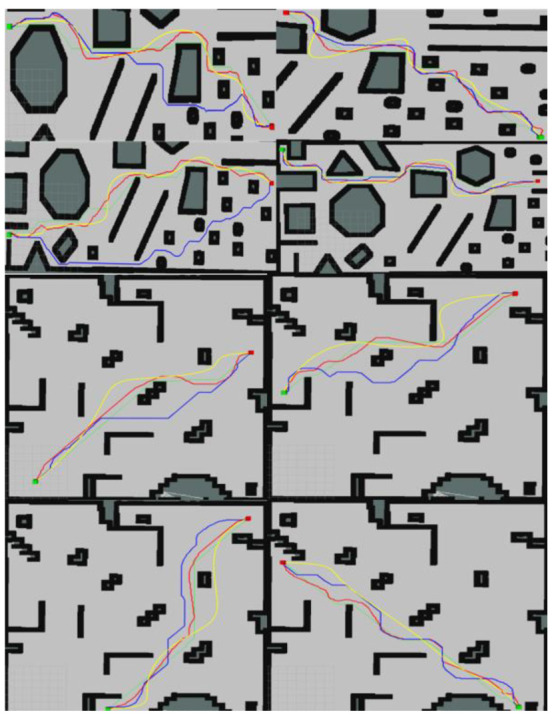
Red is the E-VSFC path, yellow is the SFC path, blue is the A*-ESDF path, and green is the origin path.

**Figure 20 sensors-25-01765-f020:**
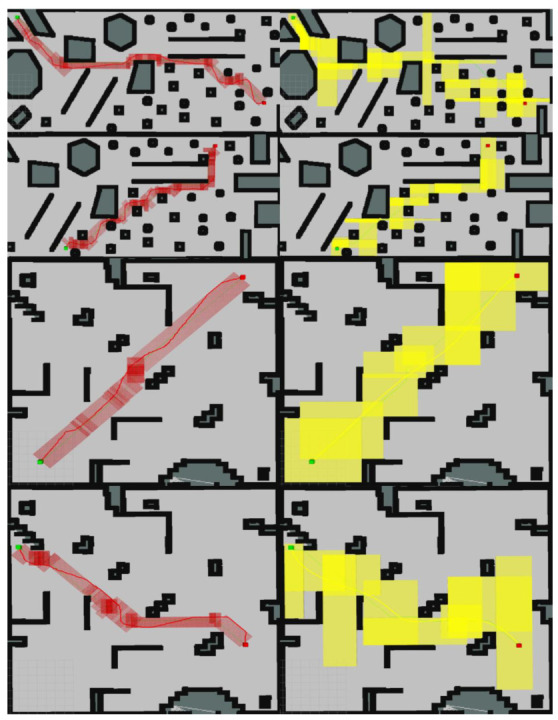
The red transparent area is the corridor constructed by E-VSFC, while the yellow transparent area is the corridor constructed by SFC. Green is start point, and red is endpoint.

**Figure 21 sensors-25-01765-f021:**
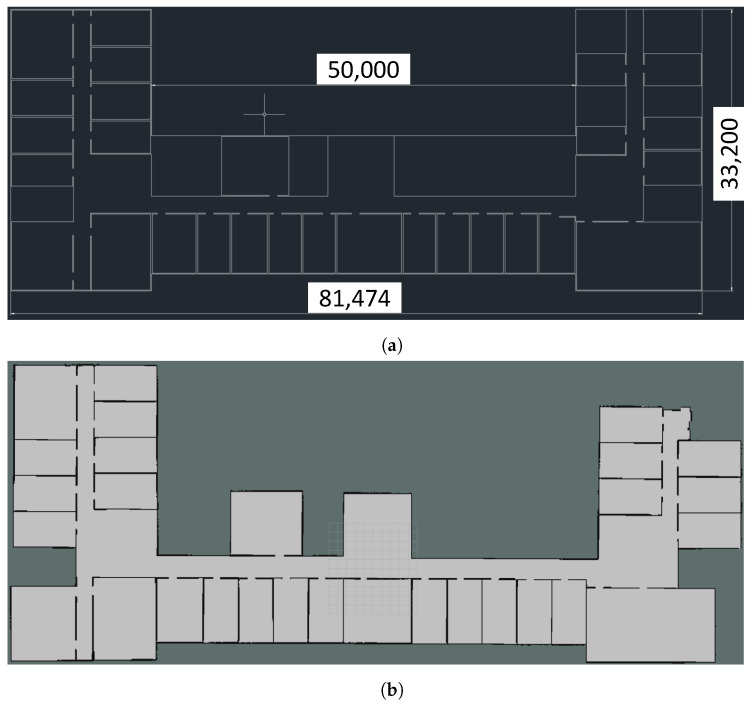
Mapping of the actual construction environment. (**a**) CAD drawing of the actual construction environment. (**b**) SLAM mapping of the actual construction environment.

**Figure 22 sensors-25-01765-f022:**
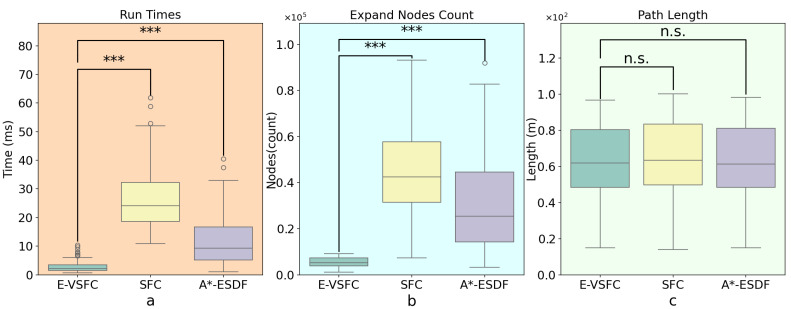
Comparison of path optimization results for E-VSFC, SFC, and A*-ESDF in the actual construction. (**a**) Comparison run times (**b**) Comparison number of expanded nodes (**c**) Comparison optimized path lengths.

**Figure 23 sensors-25-01765-f023:**
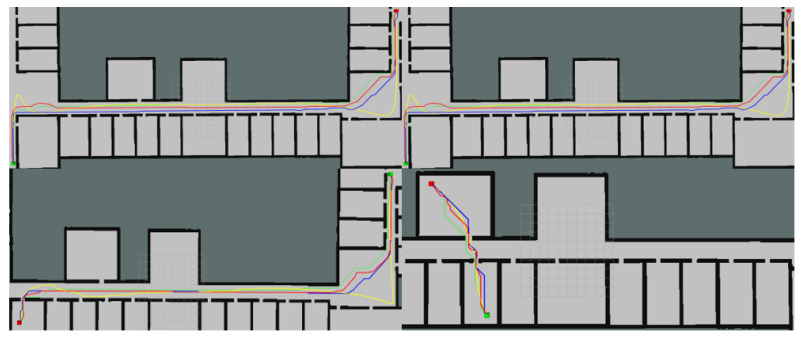
Red is E-VSFC path, yellow is SFC path, blue is A*-ESDF path, and green is origin path operational data for E-VSFC, SFC, and A*-ESDF in the actual construction site.

**Figure 24 sensors-25-01765-f024:**
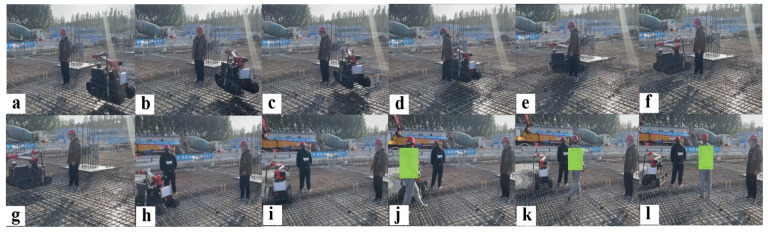
The site operation of the vibration robot: (**a**–**i**) The robot autonomously navigates, avoiding fixed obstacles; (**j**–**l**) the robot can autonomously avoid sudden obstacles during its movement.

**Table 1 sensors-25-01765-t001:** Time consumption for path sampling using vector and derivative method.

Path Points	Vector Time (ms)	Derivative Time (ms)
863	0.018	1.025
749	0.014	0.950
668	0.014	0.825
256	0.007	0.353
290	0.008	0.380
737	0.021	1.509
820	0.016	1.257
825	0.017	1.183

**Table 2 sensors-25-01765-t002:** Comparison of total path points, total ESDF values, and average ESDF values between A* and E-VSFC algorithms.

Path Points		Total ESDF		Average ESDF	
**A***	**E-VSFC**	**A***	**E-VSFC**	**A***	**E-VSFC**
498	266	37,133	16,734	75	63
283	152	21,770	8223	77	54
433	245	33,194	15,565	77	64
526	274	37,954	16,989	72	62
338	182	24,054	11,054	71	61
513	271	38,653	15,407	75	57
167	101	11,319	5346	68	53
451	263	34,915	16,183	77	62
333	188	24,739	11,407	74	61
361	200	27,911	13,495	77	67

**Table 3 sensors-25-01765-t003:** Main parameters of the chassis.

Parameter Type	Parameter Value
** *L* **	1.25 m
** *w* **	0.7 m
** $w_{mid}$ **	0.4 m
** $m_{max}$ **	193 kg
** $v_{max}$ **	±2.0 m/s
** $a_{max}$ **	±2.0 m/s^2^
** $\omega_{max}$ **	1.57 rad/s

**Table 4 sensors-25-01765-t004:** Algorithm parameters for E-VSFC, SFC, and A*-ESDF.

Algorithm	Parameter Type	Parameter Value
	Corridor Width	2.0 m
	$k_{esdf}$	2.0
E-VSFC	$k_{last}$	1.0
	$k_{end}$	1.0
	$k_{repeat}$	1.0
	Maximum Speed	±2.0 m/s
	Maximum Acceleration	±2.0 m/s^2^
SFC	Edge Distance	0.4 m
	Polynomial Degree	3
	Number Control Points	8
A*-ESDF	ESDF Coefficient	1.0

**Table 5 sensors-25-01765-t005:** Running data of E-VSFC, SFC, and A-ESDF in random maps, bold indicate the minimum values under the corresponding metrics.

Run Time (ms)			Expand Nodes			Path Length (m)		
**E-VSFC**	**SFC**	**A*-ESDF**	**E-VSFC**	**SFC**	**A*-ESDF**	**E-VSFC**	**SFC**	**A*-ESDF**
**1.04**	61.23	11.12	**4240**	28013	25807	41.28	**38.6**	43.7
**6.65**	50.79	12.15	**4814**	37594	18610	40.68	**38.7**	40.9
**2.44**	34.56	4.77	**3407**	39087	12599	31.93	**31.7**	32.9
**1.59**	48.85	3.09	**5127**	53084	8542	47.52	**46.5**	48.5
**1.37**	33.78	4.73	**4401**	51637	7099	**39.31**	39.5	40.6
**1.22**	38.62	12.21	**4939**	50474	26110	**43.82**	44.2	45.8
**1.55**	4.74	4.74	**3112**	7952	7952	25.54	**24.7**	**24.7**
**1.69**	18.75	2.45	**2594**	12777	5274	26.35	**26.3**	**26.3**
**2.79**	18.49	5.35	**3172**	13433	12462	**30.63**	31.1	32.2
**1.30**	28.52	1.50	3760	29662	**3735**	**35.56**	36.1	36.3

**Table 6 sensors-25-01765-t006:** Running data of E-VSFC, SFC, and A-ESDF in actual construction, bold indicate the minimum values under the corresponding metrics.

Run time (ms)			Expand Nodes			Path length (m)		
**E-VSFC**	**SFC**	**A*-ESDF**	**E-VSFC**	**SFC**	**A*-ESDF**	**E-VSFC**	**SFC**	**A*-ESDF**
**1.4**	23.40	4.1	**4392**	31318	13475	**58.3**	62.4	58.6
**3.3**	28.88	21.3	**6367**	52508	54811	**80.1**	82.4	80.8
**2.2**	20.88	10.6	**8818**	39697	30704	**79.1**	84.5	78.8
**1.1**	14.61	2.8	**4446**	27728	9217	44.4	46.1	**44.2**
**5.6**	52.05	18.3	**6895**	70649	51436	71.2	74.8	**70.8**
**2.2**	29.53	15.3	**7763**	64298	42951	**94.7**	97.7	95.3
**3.2**	26.08	20.9	**6796**	61511	52408	**82.4**	86.1	82.5
**1.2**	33.85	17.2	**4503**	58796	44839	**57.2**	58.2	57.7
**1.3**	24.05	8.1	**4790**	42558	23042	52.9	52.6	**52.3**
**5.0**	33.03	22.1	**7297**	53391	61409	**87.5**	89.7	88.2

**Table 7 sensors-25-01765-t007:** Statistics of planning results in the construction site.

		Stop Radius/m		Finish Task		Run Time/s		Collisions		Path Length/m	
**Static**	**Move**	**A***	**E-VSFC**	**A***	**E-VSFC**	**A***	**E-VSFC**	**A***	**E-VSFC**	**A***	**E-VSFC**
0	0	0.31	0.13	Yes	Yes	177	193	0	0	87.8	95.6
6	0	0.32	0.12	Yes	Yes	371	267	2	0	91.3	103.2
6	0	0.25	0.14	Yes	Yes	353	274	1	0	99.5	110.7
7	1	0.33	0.09	Yes	Yes	427	358	1	0	101	119.8
7	2	-	0.13	No	Yes	-	401	3	0	-	137.4
8	0	0.27	0.11	Yes	Yes	239	283	1	0	101	107.8
8	1	0.23	0.14	Yes	Yes	314	287	4	0	94.6	102.9
8	2	-	0.13	No	Yes	-	361	3	0	-	97.2
8	3	-	0.15	No	Yes	-	371	2	0	-	89.3
8	4	-	0.13	No	Yes	-	324	5	0	-	107.3

## Data Availability

The original contributions presented in this study are included in the article. Further inquiries can be directed to the corresponding author.

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
