# Peer review of "Efficient Path Planning for Collision Avoidance of Construction Vibration Robots Based on Euclidean Signed Distance Field and Vector Safety Flight Corridors"

_sensors, 2025, doi:10.3390/s25061765_

Round 1

Reviewer 1 Report

Comments and Suggestions for Authors

This paper describes a mobile manipulator (track drive) carrying a vibrator rod for removing voids from freshly poured concrete.  The paper describes the motivation for automated concrete vibrating, presents the robot for doing this, and then examines motion planning techniques for navigating the robot to drive to different locations.  The authors build upon standard A* motion planning algorithms by applying established Euclidean distance fields for finding navigation corridors and smoothed paths, which the authors have termed the "ESDF-Vector Safe Flight Corridor".   Simulation results demonstrate efficiency of the planning algorithm.  Experimental results demonstrate the robot planning motion inside an existing building and at a construction site. 

The reviewer has a number of comments:

  1. The title of the paper should be revisited.  The paper mainly focuses on path planning using existing maps of environments, whereas the title implies sensor based navigation where sensor data is used to create the map and then planning algorithms plan the motion through the map.  Likewise, the paper considers collision avoidance, where as "safe" can mean much more, such as assuring dynamically feasible paths, avoiding dangerous terrain, and considering safe distances from nearby human.  Perhaps "Efficient Path Planning for Collision Avoidance of Construction Vibration Robot  Based on ESDF and Vector Safety Flight Corridors"?
  2. The reviewer suggests that the paper might fit better into a journal like "robotics".  The focus of the paper is really about path planning and navigation, which is more of a robotics topic.  The robot does use sensors as a tool, but sensor processing/design is not really a focus of the paper.
  3. Keywords need to be revisited.  The paper does not do anything with virtual reality, medical robots, or climbing robots and should be removed.  Should "motion planning" be added?
  4. Discussion of related work in Section 2.2 could do a more thorough job of discussing efficient and safe motion planning techniques in dynamic environments with obstacles.  The section currently talks about about classical algorithms and those applied in this paper, but there is a wealth of ongoing work that should be discussed to give this paper proper context.  Just a few of the many papers to consider could include:
    1. Lee, D., Nahrendra, I.M.A., Oh, M., Yu, B., Myung, H." TRG-Planner: Traversal Risk Graph-Based Path Planning in Unstructured Environments for Safe and Efficient Navigation" IEEE Robotics and Automation Letters, 10(2), pp. 1736–1743, 2025
    2. Ren, Tianyu; Jebelli, Houtan, "Efficient 3D robotic mapping and navigation method in complex construction environments", Computer-Aided Civil and Infrastructure Engineering (2024)
    3. Yang G., Wang S., Okamura H., Ueda Y., Yasui T., Yamada T., Miyazawa Y., Yoshida S., Inada Y., Ino S., Okuhata K., Mizobuchi Y. "Safe and Efficient Motion Planning for Material Transportation Robots considering Intention Prediction of Obstacles" (2023) 2023 IEEE International Conference on Mechatronics and Automation, ICMA 2023, pp. 1179 - 1184
    4. Dai B., Khorrambakht R., Krishnamurthy P., Goncalves V., Tzes A., Khorrami F. "Safe Navigation and Obstacle Avoidance Using Differentiable Optimization Based Control Barrier Functions" (2023) IEEE Robotics and Automation Letters, 8 (9), pp. 5376 - 5383
    5. Feng K., Lu Z., Xu J., Chen H., Lou Y. "A Safety Filter for Realizing Safe Robot Navigation in Crowds" (2023) IEEE International Conference on Intelligent Robots and Systems, pp. 9729 - 9736
    6. Verginis C.K., Kantaros Y., Dimarogonas D.V. "Planning and control of multi-robot-object systems under temporal logic tasks and uncertain dynamics" (2024) Robotics and Autonomous Systems, 174, art. no. 104646
    7. Zheng A., Narayanan S.S.K.S., Vaidya U. "Safe Navigation Using Density Functions" (2023) IEEE Robotics and Automation Letters, 8 (12), pp. 8502 - 8509
    8. Park J., Kim S. "A Study on the Path Estimation of the Autonomous Mobile Robot in Extreme Environment Using Image Processing and Kalman Filter" (2023) International Conference on Control, Automation and Systems, pp. 1580 - 1583
    9. Kim Y., Chen Y., Kim S., Cho Y.K. "How Much Distance Should Robots Keep from Other Workers at Construction Jobsites?" (2024) Construction Research Congress 2024, CRC 2024, 1, pp. 893 - 902
    10. Li Z., Chen Y., Zhang Z., Zhong H., Wang Y. "FRT*: fast reactive tree for mobile robot replanning in unknown dynamic environments (2025) Robotic Intelligence and Automation.
    11. Arul S.H., Jin Park J., Prem V., Zhang Y., Manocha D. "Unconstrained Model Predictive Control for Robot Navigation under Uncertainty" (2024) Proceedings - IEEE International Conference on Robotics and Automation, pp. 9321 - 9327
    12. Rousseas P., Bechlioulis C., Kyriakopoulos K. "An Efficient Solution to Optimal Motion Planning With Provable Safety and Convergence" (2024) IEEE Open Journal of Control Systems, 3, pp. 143 - 157
    13. Tan Z., Zhang K., Shi H., Chen L., Li G. "Obstacle avoidance tracking control with antiswing and tracking errors constraint for underactuated automated lifting robots with load hoisting/lowering" (2023) Journal of Field Robotics, 40 (6), pp. 1562 - 1580
    14. Jang I., Kim H.J. "Safe Control for Navigation in Cluttered Space Using Multiple Lyapunov-Based Control Barrier Functions" (2024) IEEE Robotics and Automation Letters, 9 (3), pp. 2056 - 2063
    15. Halder S., Afsari K. "Robots in Inspection and Monitoring of Buildings and Infrastructure: A Systematic Review" (2023) Applied Sciences (Switzerland), 13 (4), art. no. 2304
    16. Wang X. Autonomous Navigation of Construction Robots Based on Visual SLAM Technology (2024) ACM International Conference Proceeding Series, pp. 139 - 143
    17. Kim Y., Yarovoi A., Han Ahn Y., Cho Y.K. "Socially Appropriate Robot Planning in Dynamic, Unseen Construction Environments" (2024) Computing in Civil Engineering 2023: Data, Sensing, and Analytics - Selected Papers from the ASCE International Conference on Computing in Civil Engineering 2023, pp. 588 - 596
    18. Calinescu R., Imrie C., Mangal R., Rodrigues G.N., Pǎsǎreanu C., Santana M.A., Vázquez G. "Controller Synthesis for Autonomous Systems with Deep-Learning Perception Components" (2024) IEEE Transactions on Software Engineering, 50 (6), pp. 1374 - 1395
    19. Yao F., Zhang T., Yuan Z., Wang Y., Chen L., Yang H. "Nonlinear coupling tracking control for underactuated construction lifting robots with load hoisting/lowering under initial input saturations" (2023) Journal of Field Robotics, 40 (2), pp. 243 - 257
    20. Huang Z., Ding K., Liu Z., Wu K., Zhang Y. A Navigation Framework Fused with 3D and 2D SLAM Algorithms for solid-state LiDARs (2023) 2023 8th IEEE International Conference on Advanced Robotics and Mechatronics, ICARM 2023, pp. 415 - 420
    21. Cloud J.M., Tram M.Q., Beksi W.J., Dupuis M.A. "Lunar Excavator Mission Operations Using Dynamic Movement Primitives" (2023) IEEE International Conference on Intelligent Robots and Systems, pp. 10708 - 10715
  5. Figure 2 left provides a screen capture of the GUI, but the labels on the buttons are in kanji characters.  Could you provide English text overlays so others can understand them?
  6. Figures and captions could be improved to indicate what is shown in them.  For example, the captions in Figs 8-11 could indicate what the different grid colors and shapes mean.
  7. Figure 11 is supposed to show an optimized verision of the path in Fig 10, but it seems to be using a different start point. Shouldn't they be the same start and end points?
  8. Equation 5 shows your technique for calculating path normals.  Numerical techniques are often sensitive to sensor noise and measurement density, which especially important with experimental data in real world applications.  Filtering data and sampling data are often applied to remedy these issues, but you do not mention these techniques in the paper.  Can you please discuss these issues and your techniques for resolving them?
  9. On line 281-282, you use a set grid size for expanding the flight corridor (i.e., "five points on one side, ten points on both sides"), but can you explain how this should this determined?  It seems to be focused on the width of the robot, but what about space required for turning due to its length?  Likewise, many researchers have focused on what envelopes are important for safe navigation around humans. 
  10. Line 349-350.  You say "…indicates a p-value far smaller than 0.001, showing an extremely significant difference", but a small p-value only shows statistical significance. The reviewer suggests toning this statement down and focusing on the effect size of the mean values a being compared. In this case, it would be relevant to talk about the importance of the ESDF values (i.e., ~62 vs ~76). 
  11. Line 381-382.  You say "Ten randomly selected sets of experimental data are shown in Table 4." Do you mean Table 5?  Table 4 indicates parameters used in the algorithms.
  12. Table 4 (P17) describes the parameters used in the algorithms.  It is not mentioned in the text and should be discussed.  More importantly, can you explain how the parameters were selected and tuned to provide the "best" results from each algorithm for comparison purposes?
  13. In figures 18-20 you provide experimental results where "the robot conducted 100 runs for algorithm testing in the actual construction environment" (quoted from Line 390).  Table 5 provides results regarding computational results using the algorithms. Can you provide information about the robot performance using these paths?  Were there any collisions with the building?  Any close calls?  How long did the robot take to execute the different paths?   Are there recommendations for tuning the parameters of your algorithm that might be helpful for others in the future?
  14. Figure 18 b, Lines 319-320. Can you indicate how was the slam map was constructed?  Was it using this robot?  If so, how was the robot navigated to create the maps?
  15. Some of the statements could be toned down a bit:
    1. Lines 61-64 you say "To ensure safe and stable operation, the robot needs a path that maintains a safe distance from obstacles. Therefore, this article deployed a safe flight corridor path optimization algorithm[12] that generates paths away from obstacles, addressing the safety issue of autonomous navigation."  The reviewer would argue that this paper addresses the issue of having a sufficiently wide corridor.  It does not address issues related to the uneven terrain or fast operations, which can cause the robot to roll over.  Nor does it address issues related to lenth of the robot and space required to turn the robot.  The reviewer suggests that the statement "addressing the safety issue of autonomous navigation" be toned down to more accurately describe the safety issues that the paper does address.
    2. For example on lines 69-71, you say "This article also introduced the Euclidean Signed Distance Field (ESDF)[13] for a comprehensive cost assessment of points within the search range, thereby accelerating the path optimization process," but [13] already did that for drones in 3D.  Maybe it would be better to say that you "applied" ESDF instead of "introduced" ESDF.
    3. In the conclusions (lines 421-424), you say "The quality and durability of concrete structures largely depend on effective concrete vibration. However, the manual vibration methods currently used on construction sites are inefficient, making it difficult to quantify and assess the quality of cement vibration. Once the cement sets, quality issues can have fatal impacts on the strength of the building, posing risks to life and property."  This article does not evaluate durability of concrete or efficiency of manual vibration techniques, however.  To draw these type of conclusions, the researchers would need to apply both manual and automated vibration and then compare the efficacy of the process and the resulting strength of the structures using destructive testing.  The reviewer recommends that the conclusion focus more on the technical contributions of this paper, which is demonstrating that they can efficiently plan paths in arbitrarily simulated environments and man-made structures.
Comments on the Quality of English Language
  1. Further editing is required:
    1. Some English needs to be improved, particularly in the beginning of the paper.  For example,
      1. "Need"  >> "needs".  As in, "…this paper needs to…"  this is in places…
      2. Nouns/Pronouns need to be added in many places.  For example, at line 128: "During the vibration

process, observed" >> "During the vibration process, WE observed…"  This is in many places.  Line 126

  1. Line 141:  "this article apply the" >> "this article applies the"
  1. Lines 88-124 has a really long paragraph.  Could you break it into two paragraphs at line 108 beginning with the sentence starting with the word "building"?
  2. The paragraph on lines 174-186 has been duplicated in this same area.
  3. Lines 201-225.  This paragraph is just rehashing much of what has already been said.  Could it shortened a bit to reduce repetition?
  4. Line 289-290.  You refer to "blue dashed lines" in Figure 9, but there are only solid lines.
  5. Line 317: "depicted in .Fig. 11." >> "depicted in Fig. 11."
  6. Lines 319-323.  Not sure what this sentence is trying to say. Is a period missing after "…Equation (6)…"?

Author Response

Q 1: The title of the paper should be revisited.  The paper mainly focuses on path planning using existing maps of environments, whereas the title implies sensor based navigation where sensor data is used to create the map and then planning algorithms plan the motion through the map.  Likewise, the paper considers collision avoidance, where as "safe" can mean much more, such as assuring dynamically feasible paths, avoiding dangerous terrain, and considering safe distances from nearby human.  Perhaps "Efficient Path Planning for Collision Avoidance of Construction Vibration Robot  Based on ESDF and Vector Safety Flight Corridors"?

A: According to your suggestion, the title of the paper has been revised and better reflects the core content of this article.

Q 2: The reviewer suggests that the paper might fit better into a journal like "robotics".  The focus of the paper is really about path planning and navigation, which is more of a robotics topic.  The robot does use sensors as a tool, but sensor processing/design is not really a focus of the paper.

A: After reviewing the available literature, we found that your journal has published several research articles on path planning algorithms similar to the topic of this paper. Therefore, we are submitting our manuscript to your journal for consideration.

Q3: Keywords need to be revisited.  The paper does not do anything with virtual reality, medical robots, or climbing robots and should be removed.  Should "motion planning" be added?

A: The original keywords "Safe and Efficient vibrating" have been deleted and replaced with "motion planning."

Q4: Discussion of related work in Section 2.2 could do a more thorough job of discussing efficient and safe motion planning techniques in dynamic environments with obstacles.  The section currently talks about about classical algorithms and those applied in this paper, but there is a wealth of ongoing work that should be discussed to give this paper proper context.  Just a few of the many papers to consider could include:

A: We have read and analyzed the examples you provided, and among them, references a, b, g, j, and t are particularly instructive for this paper. The motion planning approach for quadruped robots in complex road conditions presented in reference a could serve as a future optimization direction for our algorithm but is not applicable to the rebar vibration working environment, as the tracked chassis can adequately meet the requirements of this task. Reference b integrates LiDAR detection, SLAM, and neural networks to generate precise 3D maps. Although our working environment changes each time and does not require such an exact 3D map, this approach could provide accurate map data for future algorithm research, such as tasks in fixed environments. The density function proposed in reference g provides a convergence proof for navigation that includes safety considerations. This method is similar to ESDF but also considers the actual motion problems of the robot, making it suitable for further improvement and reference in our algorithm. The FRT* method in unknown environments described in reference j can be combined with ESDF, addressing both safe planning in unknown environments and safe obstacle avoidance in known environments based on sensor data, offering valuable ideas for future research. Finally, reference t adopts the move_base framework under the ROS robot operating system and improves the A* algorithm, providing a solid background basis for this paper.

Q5: Figure 2 left provides a screen capture of the GUI, but the labels on the buttons are in kanji characters.  Could you provide English text overlays so others can understand them?

A: The explanation of the GUI interface for the vibration robot has been added, with the figure named "Explanation of the GUI interface functions for the vibration robot." Additionally, the functions of the GUI have been introduced.

Q6: Figures and captions could be improved to indicate what is shown in them.  For example, the captions in Figs 8-11 could indicate what the different grid colors and shapes mean.

A: The meanings of different grid colors and shapes have been explained in detail in the text.

Q7: Figure 11 is supposed to show an optimized verision of the path in Fig 10, but it seems to be using a different start point. Shouldn't they be the same start and end points?

Q8: Figure 11 shows a situation that occurred during the optimization process where the algorithm did not use a comprehensive cost evaluation. Therefore, the start and end points in Figure 11 do not need to be the same as those in Figure 10. Figures 11 and 12 are presented for mutual comparison, demonstrating that the optimized path after adopting the comprehensive cost evaluation (as shown in Figure 12) is smoother.

A: To avoid repeated sampling of the original path, this paper samples the path points of the original path in the grid coordinate system. This approach ensures that the sampled path does not repeat.

Q9: On line 281-282, you use a set grid size for expanding the flight corridor (i.e., "five points on one side, ten points on both sides"), but can you explain how this should this determined?  It seems to be focused on the width of the robot, but what about space required for turning due to its length?  Likewise, many researchers have focused on what envelopes are important for safe navigation around humans.

A: The fixed grid size for expanding the flight corridor is used here to illustrate the principle of the algorithm, without considering the geometric or motion data of the robot. In the scope of this study, humans are treated as ordinary obstacles, as the obstacles have already been inflated, and the safe flight corridor is constructed based on that. Therefore, humans are not specially considered. Moreover, it is challenging for LiDAR to accurately identify humans, which would require combining visual sensors for recognition. In subsequent algorithm optimizations, the identification of humans and improvement of obstacle avoidance methods will be considered.

Q10: Line 349-350.  You say "…indicates a p-value far smaller than 0.001, showing an extremely significant difference", but a small p-value only shows statistical significance. The reviewer suggests toning this statement down and focusing on the effect size of the mean values a being compared. In this case, it would be relevant to talk about the importance of the ESDF values (i.e., ~62 vs ~76).

A: In this paper, the average ESDF value of the blank areas without obstacles was calculated for ten random maps. This represents the ESDF average value of the empty regions of the maps themselves, which is used for comparison with the average ESDF values of the paths generated by the A* algorithm and the E-VSFC-optimized paths. Additionally, a significance analysis chart of the differences in path average ESDF values for the ten random maps was redrawn: "Significance analysis of the difference in ESDF values between E-VSFC optimized path and A generated paths." The chart also includes annotations indicating the average ESDF value of the blank areas of the map for reference.

Q11: Line 381-382.  You say "Ten randomly selected sets of experimental data are shown in Table 4." Do you mean Table 5?  Table 4 indicates parameters used in the algorithms.

A: The reference to table data in the text has been revised. Table 4 shows the parameters of the algorithm, while Table 5 presents the algorithm runtime data obtained in the simulation environment.

Q12: Table 4 (P17) describes the parameters used in the algorithms.  It is not mentioned in the text and should be discussed.  More importantly, can you explain how the parameters were selected and tuned to provide the "best" results from each algorithm for comparison purposes?

A: The algorithm parameters are determined based on the geometric and motion parameters of the robot. The newly added **Table 3, "Main parameters of the chassis,"** provides the primary geometric and motion parameters of the robot. **Figure 16** is the error analysis diagram of the robot's path-following performance.

Q13: In figures 18-20 you provide experimental results where "the robot conducted 100 runs for algorithm testing in the actual construction environment" (quoted from Line 390).  Table 5 provides results regarding computational results using the algorithms. Can you provide information about the robot performance using these paths?  Were there any collisions with the building?  Any close calls?  How long did the robot take to execute the different paths?   Are there recommendations for tuning the parameters of your algorithm that might be helpful for others in the future?

A: The revised version of this paper provides the operational data of the robot in the actual environment, as shown in Table 7, "Statistics of Planning Results in the Construction Site." This table mainly presents the performance of the A* algorithm and the E-VSFC algorithm under different numbers of static and dynamic obstacles. The data includes parking radius, whether the vibration compaction task was completed, the actual runtime of the robot, whether any collisions occurred during operation, and the length of the robot's travel path.  For future suggestions on tuning the algorithm parameters: focusing on path smoothness can increase the values of ; emphasizing safety can enhance the value of ; and reducing energy consumption can improve the values of  and . These recommendations may be helpful for others in the future.

Q14: Figure 18 b, Lines 319-320. Can you indicate how was the slam map was constructed?  Was it using this robot?  If so, how was the robot navigated to create the maps?

A: The hardware uses the Mid360 LiDAR and BMI088 sensor, while the software employs the Fast-lio2 algorithm. We process the collected PCD data by layering it according to the method shown in Fig. 4 and then merge it to construct the map. The PCD data collection is completed during the process of the robot performing its navigation task.

Q15: Some of the statements could be toned down a bit:

A: The content of the mentioned sections has been revised, and the language throughout the entire article has been refined with detailed adjustments. This includes removing repetitive parts, simplifying verbose expressions, and rephrasing unclear statements for better clarity.

Q: Comments on the Quality of English Language

The manuscript has been revised according to all the suggestions provided: 

- "Need" has been changed to "needs." 

- Nouns or pronouns have been added in places such as "During the vibration process, observed," which now reads correctly with proper subjects. 

- "This article apply the" has been corrected to "this article applies the." 

- The long paragraph from lines 88–124 in the original draft has been split at line 108, starting with the sentence beginning with "Building." 

- The duplicated content in lines 174–186 of the original draft has been removed. 

- The paragraph from lines 201–225 in the original draft has been shortened and streamlined for clarity. 

- The inaccurate description of Figure 9 has been corrected. 

- "Depicted in .Fig. 11." has been revised to "depicted in Fig. 11." 

- A period has been added in the sentence at lines 319–323 of the original draft where needed.

Reviewer 2 Report

Comments and Suggestions for Authors

The article addresses a crucial topic: advancing automation in concrete work and its implications for robotics and civil engineering. Implementing automated solutions through robots could improve safety, enhance working conditions, and reduce the need for professionals to address issues such as cracks. This study introduces a method for optimizing automated path safety for vibrating robots in concrete vibration tasks, utilizing safe flight corridors and Euclidean signed distance fields.

While the paper is well-written and structured, I believe the following improvements should be made before it can be accepted:

1. Although the authors mention that the algorithm is very efficient in dynamic environments, none of the simulations showcase results involving moving obstacles, such as workers. It would be beneficial to include simulations with moving obstacles.

2. For the point above, a statistical analysis is necessary to demonstrate the algorithm's efficiency in highly crowded environments. Additionally, it should outline the conditions under which the algorithm may fail—specifically, how many moving and static obstacles can be present before the robot cannot reach the vibration point; what is the failure rate compared with other methods, e.g. A*?
5. Incorporating a video showcasing the robot in action during actual testing will greatly enhance the reader’s experience.

4. The paper needs careful proofreading.

Author Response

Q1: Although the authors mention that the algorithm is very efficient in dynamic environments, none of the simulations showcase results involving moving obstacles, such as workers. It would be beneficial to include simulations with moving obstacles.

A: The revised version of this paper provides the operational data of the robot in real-world environments in Table 7, "Statistics of Planning Results in the Construction Site." This table primarily shows the performance of the A* algorithm and the E-VSFC algorithm under varying numbers of static and dynamic obstacles. The data includes parking radius, whether the compaction task was completed, actual robot runtime, whether any collisions occurred during operation, and the length of the robot's travel path. These results are from real-world tests rather than simulations.

Q2: For the point above, a statistical analysis is necessary to demonstrate the algorithm's efficiency in highly crowded environments. Additionally, it should outline the conditions under which the algorithm may fail—specifically, how many moving and static obstacles can be present before the robot cannot reach the vibration point; what is the failure rate compared with other methods, e.g. A*?

A: In response to Question 1, **Table 7, "Statistics of Planning Results in the Construction Site,"** provides statistics on the robot's collision scenarios under varying numbers of static and dynamic obstacles. Additionally, it shows that as the number of obstacles increases, the task completion rate decreases when using the A* algorithm, while the runtime for the E-VSFC algorithm increases but ensures successful navigation task completion.

Q3: Incorporating a video showcasing the robot in action during actual testing will greatly enhance the reader’s experience.

A: Figure 23 shows the operation of the robot navigating and automatically avoiding obstacles. We can provide a video of the robot's operation; however, due to the requirements of the client, we cannot publish the video on public video platforms. If you are interested, please contact the email address: 2410101@stu.neu.edu.cn. After obtaining the client's consent and signing an agreement, we can provide the video to you.

Q4: The paper needs careful proofreading.

The introduction section of the paper has been revised. Throughout the paper, issues such as repetition and unclear expressions have been addressed and improved. Additionally, charts and graphs have been added in the section validating the algorithm's effectiveness to better illustrate the algorithm's operational results.

Reviewer 3 Report

Comments and Suggestions for Authors

The research is devoted to the development and testing of algorithms for path planning of the mobile robot. The robot is designed to perform a technological operation during the construction of buildings (vibration of poured concrete). The track chassis moves the robot to a pre-specified position, and the manipulator arm positions the vibrating tool in a layer of freshly poured concrete. The path to the destination point must be planned taking into account obstacles.
A comparison of three algorithms for constructing a path is carried out: E-VSFC, SFC, A*-ESDF. It is shown that E-VSFC is optimal compared to the others (three criteria were studied: run time, path length, accessed nodes).
The algorithms have been tested in simulations and field experiments.
The results of the study can be used to automate construction work during the construction of monolithic buildings.
Note that the proposed algorithms can be used for any mobile robots moving in an environment with obstacles.
The article is well organized: a detailed overview of the sources is provided, a description and purpose of the robot are given, a detailed description of path formation algorithms is given, a comparative analysis of modeling results is provided, illustrations, graphs and tables reflect the research results well. The conclusions are consistent with the task at hand.
Let's make a few remarks.

1. When planning a path, the maneuverability characteristics of the robot's chassis are not taken into account: maximum speed and acceleration affect, for example, the permissible curvature of the path.
2. The accuracy of the robot's movement along the trajectory is not specified. This parameter must be taken into account in the permissible distance to obstacles.

Author Response

Q1: When planning a path, the maneuverability characteristics of the robot's chassis are not taken into account: maximum speed and acceleration affect, for example, the permissible curvature of the path.

A: The motion parameters of the robot, including maximum speed and acceleration, as well as the geometric parameters of the robot (see Table 3), have been added to the manuscript. These parameters have been verified using Equation (7) and Figure 16 to ensure that the corridor width can guarantee the robot's safety. Since the robot uses a differential drive chassis, it is capable of on-the-spot rotation, thus requiring less stringent path curvature constraints.

Q2: The accuracy of the robot's movement along the trajectory is not specified. This parameter must be taken into account in the permissible distance to obstacles.

A: The motion accuracy of the robot along the trajectory has been provided in the second paragraph of Section 4.3.1. It is verified through Figure 16, "Tracked chassis for vibration robots," and Equation (8) that the corridor width presented in the paper satisfies the safety requirements.

Round 2

Reviewer 1 Report

Comments and Suggestions for Authors

The revised paper addresses most comments, but there are some remaining concerns and new concerns to address:

  1. Line 337:  You say "the red circle area of Fig. 11, making"  but there does not appear be a red circle in fig 11. 
  2. Line 370-371: You say "This article use the average Euclidean Signed Distance 370 Field (ESDF) value of the path points as a safety measure; the smaller the average ESDF value of the path points, the safer the path is considered. Can you specify how exactly the ESDF value is calculated and how lower values naturally lead to safeer paths?
  3. Line 383: You say "The average ESDF value of the the E-VSFC-optimized paths was 23% lower than that of the A* paths".  Can you say a little more about what this means?
  4. Line 388:  In reference to Figure 14, you say "The symbol "***" indicates that the p-value is much less than 0.001, suggesting an extremely significant difference" but this only indicates that normal distributions fit to the data have minimal overlap.  It would be better to say that they are "statistically different".  The effect size (23%) mentioned and the fact that it is statiscally significant is what matters.
  5. While the statistical analysis and effect size indicate that a difference is noted on a average, Figure 14 also indicates some of the E-VSFC outliers have ESDF values that are nearly as large as the the nominal A* results. Likewise, some of the whiskers are clearly overlapping witgh some of the maps even though the boxes do not overlap.  Similar results happen with some of the A* results.  Can you say what happened in these cases and what might be done to resolve these issues?
  6. Figure 21(a) highligts similar outliers with the E-VSFC run time compared to the A*-ESDF. While the stastics indicate that they are statistically different, there are outliers that suggest in some cases the proposed E-VSFC algorithm takes much longer.  Why is that in these cases? 
  7. Line 389:  You then say "This demonstrates that the E-VSFC algorithm is effective in safely optimizing the original paths," but how do the ESDF values reported in Figure 14 indicate this?  

Comments on the Quality of English Language

The writing revisions look good, but a few things were noticed…

  1. Line 76:  "Named this algorithm the ESDF-Vector Safe Flight Corridor (E-VSFC)."  is missing a subject.  Maybe "We named this algorithm…"
  2. Line 420:  "opreate" >> "operate"

Author Response

Q1: Line 337:  You say "the red circle area of Fig. 11, making"  but there does not appear be a red circle in fig 11.

A: I have modified Figure 11, and the red circle has been added to the diagram. The area circled in red highlights a section where the path makes a significant turn.

Q2: Line 370-371: You say "This article use the average Euclidean Signed Distance 370 Field (ESDF) value of the path points as a safety measure; the smaller the average ESDF value of the path points, the safer the path is considered. Can you specify how exactly the ESDF value is calculated and how lower values naturally lead to safer paths?

A: In the text, Equations (7) and (8) have been added to illustrate the calculation method of the Euclidean Signed Distance Field (ESDF). A higher ESDF value indicates that the current path point is closer to the nearest obstacle, which may result in a collision with the obstacle due to control errors. In contrast, a lower ESDF value indicates that the current path point is farther from the nearest obstacle, thus being safer.

Q3: Line 383: You say "The average ESDF value of the the E-VSFC-optimized paths was 23% lower than that of the A* paths".  Can you say a little more about what this means?

A: Combining the response to Q2 and the findings in Figure 14, where the average ESDF value of the E-VSFC optimized paths is lower than the map's average ESDF value, and the average ESDF value of the original A* generated paths is higher than the map's average ESDF value, it can be concluded that the path points optimized by E-VSFC are farther from obstacles and therefore safer.

Q4: Line 388:  In reference to Figure 14, you say "The symbol "***" indicates that the p-value is much less than 0.001, suggesting an extremely significant difference" but this only indicates that normal distributions fit to the data have minimal overlap.  It would be better to say that they are "statistically different".  The effect size (23%) mentioned and the fact that it is statiscally significant is what matters.

A: Here's the translation of your revised statement with the added context and explanation regarding Figure 14 to demonstrate the safety optimization effect of E-VSFC.

Q5: While the statistical analysis and effect size indicate that a difference is noted on a average, Figure 14 also indicates some of the E-VSFC outliers have ESDF values that are nearly as large as the the nominal A* results. Likewise, some of the whiskers are clearly overlapping witgh some of the maps even though the boxes do not overlap.  Similar results happen with some of the A* results.  Can you say what happened in these cases and what might be done to resolve these issues?

A: Added Figure 15 (The situation where the initial path generates a very narrow corridor in densely obstructed areas, resulting in minimal optimization effects) to illustrate that the outliers in E-VSFC are comparable to the A* results and to highlight the issues with the discrepancy test in Figure 14. Referring to Figure 15, where the start and end points are randomly selected, it shows that when the initial path is situated amidst obstacles and generates a very narrow corridor, the scenario depicted in Figure 14 occurs. The E-VSFC optimized path is based on the original path; to solve this issue, improvements must be made to the initial path. The E-VSFC algorithm performs safety optimizations based on the initial path without altering the original trajectory.

Q6: Figure 21(a) highligts similar outliers with the E-VSFC run time compared to the A*-ESDF. While the stastics indicate that they are statistically different, there are outliers that suggest in some cases the proposed E-VSFC algorithm takes much longer.  Why is that in these cases?

A: Because Figure 15 was added to address Q5 and to improve the article, the original Figure 21 is now Figure 22. The runtime length of the E-VSFC algorithm depends on the length of the original path; the longer the original path, the longer the E-VSFC optimization time. Combining the data in Table 6, it can be seen that as the path length increases, the run times of both E-VSFC and A*-ESDF increase. Under the same start and end points, the runtime of E-VSFC is shorter than that of A*-ESDF. In some cases, the E-VSFC algorithm takes more time because the start and end points are different, leading to such situations.

Q7: Line 389:  You then say "This demonstrates that the E-VSFC algorithm is effective in safely optimizing the original paths," but how do the ESDF values reported in Figure 14 indicate this? 

A: In Figure 14, the average ESDF value of the A* paths is higher than the map's average ESDF value, while the average ESDF value of the E-VSFC optimized paths is lower than the map's average ESDF value. Additionally, the data in Table 2 shows that the mean ESDF value of the E-VSFC path points is 23% lower than that of the A* paths. Collectively, these findings indicate that the E-VSFC algorithm is effective in safely optimizing the original paths.

Q8: Line 76:  "Named this algorithm the ESDF-Vector Safe Flight Corridor (E-VSFC)."  is missing a subject.  Maybe "We named this algorithm…"

A: We have corrected the error in the original text by adding 'We' before 'Named this algorithm the ESDF-Vector Safe Flight Corridor (E-VSFC).'

Q9: Line 420:  "opreate" >> "operate"

A: Already modified. Thank you to the reviewer for their comments. We have also corrected similar errors throughout the document.

Reviewer 2 Report

Comments and Suggestions for Authors

The authors answered all of the questions and I think the paper is now ready to be accepted. 

Author Response

Q: The authors answered all of the questions and I think the paper is now ready to be accepted.

A: We have addressed all the questions raised by the reviewer and made detailed revisions to the article. Thank you for the reviewer's valuable feedback.